# Robust Reinforcement Learning using Offline Data

**Kishan Panaganti[1], Zaiyan Xu[1], Dileep Kalathil[1], Mohammad Ghavamzadeh[2]**
[1]Texas A&M University, [2]Google Research.
Emails: {kpb, zxu43, dileep.kalathil}@tamu.edu, ghavamza@google.com

## Abstract

The goal of robust reinforcement learning (RL) is to learn a policy that is robust against the uncertainty in model parameters. Parameter uncertainty commonly occurs in many real-world RL applications due to simulator modeling errors, changes in the real-world system dynamics over time, and adversarial disturbances. Robust RL is typically formulated as a max-min problem, where the objective is to learn the policy that maximizes the value against the worst possible models that lie in an uncertainty set. In this work, we propose a robust RL algorithm called Robust Fitted Q-Iteration (RFQI), which uses only an offline dataset to learn the optimal robust policy. Robust RL with offline data is significantly more challenging than its non-robust counterpart because of the minimization over all models present in the robust Bellman operator. This poses challenges in offline data collection, optimization over the models, and unbiased estimation. In this work, we propose a systematic approach to overcome these challenges, resulting in our RFQI algorithm. We prove that RFQI learns a near-optimal robust policy under standard assumptions and demonstrate its superior performance on standard benchmark problems.

## 1 Introduction

Reinforcement learning (RL) algorithms often require a large number of data samples to learn a control policy. As a result, training them directly on the real-world systems is expensive and potentially dangerous. This problem is typically overcome by training them on a simulator (online RL) or using a pre-collected offline dataset (offline RL). The offline dataset is usually collected either from a sophisticated simulator of the real-world system or from the historical measurements. The trained RL policy is then deployed assuming that the training environment, the simulator or the offline data, faithfully represents the model of the real-world system. This assumption is often incorrect due to multiple factors such as the approximation errors incurred while modeling, changes in the real-world parameters over time and possible adversarial disturbances in the real-world. For example, the standard simulator settings of the sensor noise, action delay, friction, and mass of a mobile robot can be different from that of the actual real-world robot, in addition to changes in the terrain, weather conditions, lighting, and obstacle densities of the testing environment. Unfortunately, the current RL control policies can fail dramatically when faced with even mild changes in the training and testing environments (Sünderhauf et al., 2018; Tobin et al., 2017; Peng et al., 2018).

The goal in robust RL is to learn a policy that is robust against the model parameter mismatches between the training and testing environments. The robust planning problem is formalized using the framework of Robust Markov Decision Process (RMDP) (Iyengar, 2005; Nilim and El Ghaoui, 2005). Unlike the standard MDP which considers a single model (transition probability function), the RMDP formulation considers a set of models which is called the *uncertainty set*. The goal is to find an optimal robust policy that performs the best under the worst possible model in this uncertainty set. The minimization over the uncertainty set makes the robust MDP and robust RL problems significantly more challenging than their non-robust counterparts.

36th Conference on Neural Information Processing Systems (NeurIPS 2022).

In this work, we study the problem of developing a robust RL algorithm with provably optimal performance for an RMDP with arbitrarily large state spaces, using only offline data with function approximation. Before stating the contributions of our work, we provide a brief overview of the results in offline and robust RL that are directly related to ours. We leave a more thorough discussion on related works to Appendix D.

**Offline RL:** Offline RL considers the problem of learning the optimal policy only using a pre-collected (offline) dataset. Offline RL problem has been addressed extensively in the literature (Antos et al., 2008; Bertsekas, 2011; Lange et al., 2012; Chen and Jiang, 2019; Xie and Jiang, 2020; Levine et al., 2020; Xie et al., 2021). Many recent works develop deep RL algorithms and heuristics for the offline RL problem, focusing on the algorithmic and empirical aspects (Fujimoto et al., 2019; Kumar et al., 2019, 2020; Yu et al., 2020; Zhang and Jiang, 2021). A number of theoretical work focus on analyzing the variations of Fitted Q-Iteration (FQI) algorithm (Gordon, 1995; Ernst et al., 2005), by identifying the necessary and sufficient conditions for the learned policy to be approximately optimal and characterizing the performance in terms of sample complexity (Munos and Szepesvári, 2008; Farahmand et al., 2010; Lazaric et al., 2012; Chen and Jiang, 2019; Liu et al., 2020; Xie et al., 2021). All these works assume that the offline data is generated according to a single model and the goal is to find the optimal policy for the MDP with the same model. In particular, none of these works consider the *offline robust RL problem* where the offline data is generated according to a (training) model which can be different from the one in testing, and the goal is to learn a policy that is robust w.r.t. an uncertainty set.

**Robust RL:** The RMDP framework was first introduced in Iyengar (2005); Nilim and El Ghaoui (2005). The RMDP problem has been analyzed extensively in the literature (Xu and Mannor, 2010; Wiesemann et al., 2013; Yu and Xu, 2015; Mannor et al., 2016; Russel and Petrik, 2019) providing computationally efficient algorithms, but these works are limited to the planning problem. Robust RL algorithms with provable guarantees have also been proposed (Lim et al., 2013; Tamar et al., 2014; Roy et al., 2017; Panaganti and Kalathil, 2021; Wang and Zou, 2021), but they are limited to tabular or linear function approximation settings and only provide asymptotic convergence guarantees. Robust RL problem has also been addressed using deep RL methods (Pinto et al., 2017; Derman et al., 2018, 2020; Mankowitz et al., 2020; Zhang et al., 2020a). However, these works do not provide any theoretical guarantees on the performance of the learned policies.

The works that are closest to ours are by Zhou et al. (2021); Yang et al. (2021); Panaganti and Kalathil (2022) that address the robust RL problem in a tabular setting under the generative model assumption. Due to the generative model assumption, the offline data has the same uniform number of samples corresponding to each and every state-action pair, and tabular setting allows the estimation of the uncertainty set followed by solving the planning problem. Our work is significantly different from these in the following way: $(i)$ we consider a robust RL problem with arbitrary large state space, instead of the small tabular setting, $(ii)$ we consider a true offline RL setting where the state-action pairs are sampled according to an arbitrary distribution, instead of using the generative model assumption, $(iii)$ we focus on a function approximation approach where the goal is to directly learn optimal robust value/policy using function approximation techniques, instead of solving the tabular planning problem with the estimated model. *To the best of our knowledge, this is the first work that addresses the offline robust RL problem with arbitrary large state space using function approximation, with provable guarantees on the performance of the learned policy.*

**Offline Robust RL: Challenges and Our Contributions:** Offline robust RL is significantly more challenging than its non-robust counterpart mainly because of the following key difficulties.
$(i)$ Data generation: The optimal robust policy is computed by taking the infimum over all models in the uncertainty set $\mathcal{P}$. However, generating data according to all models in $\mathcal{P}$ is clearly infeasible. It may only be possible to get the data from a nominal (training) model $P^o$. *How do we use the data from a nominal model to account for the behavior of all the models in the uncertainty set $\mathcal{P}$?*
$(ii)$ Optimization over the uncertainty set $\mathcal{P}$: The robust Bellman operator (defined in (3)) involves a minimization over $\mathcal{P}$, which is a significant computational challenge. Moreover, the uncertainty set $\mathcal{P}$ itself is unknown in the RL setting. *How do we solve the optimization over $\mathcal{P}$?*
$(iii)$ Function approximation: Approximation of the robust Bellman update requires a modified target function which also depends on the approximate solution of the optimization over the uncertainty set. *How do we perform the offline RL update accounting for both approximations?*

As the *key technical contributions* of this work, we first derive a dual reformulation of the robust Bellman operator which replaces the expectation w.r.t. all models in the uncertainty set $\mathcal{P}$ with an ex-

pectation only w.r.t. the nominal (training) model $P^o$. This enables using the offline data generated by $P^o$ for learning, without relying on high variance importance sampling techniques to account for all models in $\mathcal{P}$. Following the same reformulation, we then show that the optimization problem over $\mathcal{P}$ can be further reformulated as functional optimization. We solve this functional optimization problem using empirical risk minimization and obtain performance guarantees using the Rademacher complexity based bounds. We then use the approximate solution obtained from the empirical risk minimization to generate modified target samples that are then used to approximate robust Bellman update through a generalized least squares approach with provably bounded errors. Performing these operations iteratively results in our proposed Robust Fitted Q-Iteration (RFQI) algorithm, for which we prove that its learned policy achieves non-asymptotic and approximately optimal performance guarantees.

**Notations:** For a set $\mathcal{X}$, we denote its cardinality as $|\mathcal{X}|$. The set of probability distribution over $\mathcal{X}$ is denoted as $\Delta(\mathcal{X})$, and its power set sigma algebra as $\Sigma(\mathcal{X})$. For any $x \in \mathbb{R}$, we denote $\max\{x, 0\}$ as $(x)_+$. For any function $f : \mathcal{S} \times \mathcal{A} \to \mathbb{R}$, state-action distribution $\nu \in \Delta(\mathcal{S} \times \mathcal{A})$, and real number $p \geq 1$, the $\nu$-weighted $p$-norm of $f$ is defined as $\|f\|_{p,\nu} = \mathbb{E}_{s,a \sim \nu}[|f(s,a)|^p]^{1/p}$.

## 2  Preliminaries

A Markov Decision Process (MDP) is a tuple $(\mathcal{S}, \mathcal{A}, r, P, \gamma, d_0)$, where $\mathcal{S}$ is the state space, $\mathcal{A}$ is the action space, $r : \mathcal{S} \times \mathcal{A} \to \mathbb{R}$ is the reward function, $\gamma \in (0, 1)$ is the discount factor, and $d_0 \in \Delta(\mathcal{S})$ is the initial state distribution. The transition probability function $P_{s,a}(s')$ is the probability of transitioning to state $s'$ when action $a$ is taken at state $s$. In the literature, $P$ is also called the *model* of the MDP. We consider a setting where $|\mathcal{S}|$ and $|\mathcal{A}|$ are finite but can be arbitrarily large. We will also assume that $r(s, a) \in [0, 1]$, for all $(s, a) \in \mathcal{S} \times \mathcal{A}$, without loss of generality. A policy $\pi : \mathcal{S} \to \Delta(\mathcal{A})$ is a conditional distribution over actions given a state. The value function $V_{\pi,P}$ and the state-action value function $Q_{\pi,P}$ of a policy $\pi$ for an MDP with model $P$ are defined as

$$V_{\pi,P}(s) = \mathbb{E}_{\pi,P}\Big[\sum_{t=0}^{\infty} \gamma^t r(s_t, a_t) \mid s_0 = s\Big], \quad Q_{\pi,P}(s,a) = \mathbb{E}_{\pi,P}\Big[\sum_{t=0}^{\infty} \gamma^t r(s_t, a_t) \mid s_0 = s, a_0 = a\Big],$$

where the expectation is over the randomness induced by the policy $\pi$ and model $P$. The optimal value function $V_P^*$ and the optimal policy $\pi_P^*$ of an MDP with the model $P$ are defined as $V_P^* = \max_\pi V_{\pi,P}$ and $\pi_P^* = \arg\max_\pi V_{\pi,P}$. The optimal state-action value function is given by $Q_P^* = \max_\pi Q_{\pi,P}$. The optimal policy can be obtained as $\pi_P^*(s) = \arg\max_a Q_P^*(s,a)$. The discounted state-action occupancy of a policy $\pi$ for an MDP with model $P$, denoted as $d_{\pi,P} \in \Delta(\mathcal{S} \times \mathcal{A})$, is defined as $d_{\pi,P}(s,a) = (1-\gamma)\mathbb{E}_{\pi,P}[\sum_{t=0}^{\infty} \gamma^t \mathbb{1}(s_t = s, a_t = a)]$.

**Robust Markov Decision Process (RMDP):** Unlike the standard MDP which considers a single model (transition probability function), the RMDP formulation considers a set of models. We refer to this set as the *uncertainty set* and denote it as $\mathcal{P}$. We consider $\mathcal{P}$ that satisfies the standard $(s, a)$-*rectangularity condition* (Iyengar, 2005). We note that a similar uncertainty set can be considered for the reward function at the expense of additional notations. However, since the analysis will be similar and the sample complexity guarantee will be identical up to a constant factor, without loss of generality, we assume that the reward function is known and deterministic.

We specify an RMDP as $M = (\mathcal{S}, \mathcal{A}, r, \mathcal{P}, \gamma, d_0)$, where the uncertainty set $\mathcal{P}$ is typically defined as

$$\mathcal{P} = \otimes_{(s,a) \in \mathcal{S} \times \mathcal{A}} \mathcal{P}_{s,a}, \quad \text{where } \mathcal{P}_{s,a} = \{P_{s,a} \in \Delta(\mathcal{S}) \; : \; D(P_{s,a}, P_{s,a}^o) \leq \rho\}, \tag{1}$$

$P^o = (P_{s,a}^o, (s, a) \in \mathcal{S} \times \mathcal{A})$ is the *nominal model*, $D(\cdot, \cdot)$ is a distance metric between two probability distributions, and $\rho > 0$ is the radius of the uncertainty set that indicates the level of robustness. The nominal model $P^o$ can be thought as the model of the training environment. It is either the model of the simulator on which the (online) RL algorithm is trained, or in our setting, it is the model according to which the offline data is generated. The uncertainty set $\mathcal{P}$ (1) is the set of all valid transition probability functions (valid testing models) in the neighborhood of the nominal model $P^o$, which by definition satisfies $(s, a)$-rectangularity condition (Iyengar, 2005), where the neighborhood is defined using the distance metric $D(\cdot, \cdot)$ and radius $\rho$. In this work, we consider the *Total Variation (TV) uncertainty set* defined using the TV distance, i.e., $D(P_{s,a}, P_{s,a}^o) = (1/2)\|P_{s,a} - P_{s,a}^o\|_1$.

The RMDP problem is to find the optimal robust policy which maximizes the value against the worst possible model in the uncertainty set $\mathcal{P}$. The *robust value function* $V^\pi$ corresponding to a policy $\pi$

and the *optimal robust value function $V^*$* are defined as (Iyengar, 2005; Nilim and El Ghaoui, 2005)

$$V^\pi = \inf_{P \in \mathcal{P}} V_{\pi,P}, \qquad V^* = \sup_\pi \inf_{P \in \mathcal{P}} V_{\pi,P}. \tag{2}$$

The *optimal robust policy $\pi^*$* is such that the robust value function corresponding to it matches the optimal robust value function, i.e., $V^{\pi^*} = V^*$. It is known that there exists a deterministic optimal policy (Iyengar, 2005) for the RMDP. The *robust Bellman operator* is defined as (Iyengar, 2005)

$$(TQ)(s,a) = r(s,a) + \gamma \inf_{P_{s,a} \in \mathcal{P}_{s,a}} \mathbb{E}_{s' \sim P_{s,a}}[\max_b Q(s',b)]. \tag{3}$$

It is known that $T$ is a contraction mapping in the infinity norm and hence it has a unique fixed point $Q^*$ with $V^*(s) = \max_a Q^*(s,a)$ and $\pi^*(s) = \arg\max_a Q^*(s,a)$ (Iyengar, 2005). The *Robust Q-Iteration (RQI)* can now be defined using the robust Bellman operator as $Q_{k+1} = TQ_k$. Since $T$ is a contraction, it follows that $Q_k \to Q^*$. So, RQI can be used to compute (solving the planning problem) $Q^*$ and $\pi^*$ in the tabular setting with a known $\mathcal{P}$. Due to the optimization over the uncertainty set $\mathcal{P}_{s,a}$ for each $(s,a)$ pair, solving the planning problem in RMDP using RQI is much more computationally intensive than solving it in MDP using Q-Iteration.

**Offline RL:** Offline RL considers the problem of learning the optimal policy of an MDP when the algorithm does not have direct access to the environment and cannot generate data samples in an online manner. For learning the optimal policy $\pi_P^*$ of an MDP with model $P$, the algorithm will only have access to an offline dataset $\mathcal{D}_P = \{(s_i, a_i, r_i, s_i')\}_{i=1}^N$, where $(s_i, a_i) \sim \mu$, $\mu \in \Delta(\mathcal{S} \times \mathcal{A})$ is some distribution, and $s_i' \sim P_{s_i,a_i}$. *Fitted Q-Iteration (FQI)* is a popular offline RL approach which is amenable to theoretical analysis while achieving impressive empirical performance. In addition to the dataset $\mathcal{D}_P$, FQI uses a function class $\mathcal{F} = \{f : \mathcal{S} \times \mathcal{A} \to [0, 1/(1-\gamma)]\}$ to approximate $Q_P^*$. The typical FQI update is given by $f_{k+1} = \arg\min_{f \in \mathcal{F}} \sum_{i=1}^N (r(s_i, a_i) + \gamma \max_b f_k(s_i', b) - f(s_i, a_i))^2$, which aims to approximate the non-robust Bellman update using offline data with function approximation. Under suitable assumptions, it is possible to obtain provable performance guarantees for FQI (Szepesvári and Munos, 2005; Chen and Jiang, 2019; Liu et al., 2020).

## 3 Offline Robust Reinforcement Learning

The goal of an offline robust RL algorithm is to learn the optimal robust policy $\pi^*$ using a pre-collected offline dataset $\mathcal{D}$. The data is typically generated according to a nominal (training) model $P^o$, i.e., $\mathcal{D} = \{(s_i, a_i, r_i, s_i')\}_{i=1}^N$, where $(s_i, a_i) \sim \mu, \mu \in \Delta(\mathcal{S} \times \mathcal{A})$ is some data generating distribution, and $s_i' \sim P_{s_i,a_i}^o$. The uncertainty set $\mathcal{P}$ is defined around this nominal model $P^o$ as given in (1) w.r.t. the total variation distance metric. We emphasize that the learning algorithm does not know the nominal model $P^o$ as it has only access to $\mathcal{D}$, and hence it also does not know $\mathcal{P}$. Moreover, the learning algorithm does not have data generated according to any other models in $\mathcal{P}$ and has to rely only on $\mathcal{D}$ to account for the behavior w.r.t. all models in $\mathcal{P}$.

Learning policies for RL problems with large state-action spaces is computationally intractable. RL algorithms typically overcome this issue by using function approximation. In this paper, we consider two function classes $\mathcal{F} = \{f : \mathcal{S} \times \mathcal{A} \to [0, 1/(1-\gamma)]\}$ and $\mathcal{G} = \{g : \mathcal{S} \times \mathcal{A} \to [0, 2/(\rho(1-\gamma))]\}$. We use $\mathcal{F}$ to approximate $Q^*$ and $\mathcal{G}$ to approximate the dual variable functions which we will introduce in the next section. For simplicity, we will first assume that these function classes are finite but exponentially large, and we will use the standard log-cardinality to characterize the sample complexity results, as given in Theorem 1. We note that, at the cost of additional notations and analysis, infinite function classes can also be considered where the log-cardinalities are replaced by the appropriate notions of covering number.

Similar to the non-robust offline RL, we make the following standard assumptions about the data generating distribution $\mu$ and the representation power of $\mathcal{F}$.

**Assumption 1** (Concentratability). *There exists a finite constant $C > 0$ such that for any $\nu \in \{d_{\pi,P^o} \mid any\ policy\ \pi\} \subseteq \Delta(\mathcal{S} \times \mathcal{A})$, we have $\|\nu/\mu\|_\infty \le \sqrt{C}$.*

Assumption 1 states that the ratio of the distribution $\nu$ and the data generating distribution $\mu$, $\nu(s,a)/\mu(s,a)$, is uniformly bounded. This assumption is widely used in the offline RL literature (Munos, 2003; Agarwal et al., 2019; Chen and Jiang, 2019; Wang et al., 2021; Xie et al., 2021) in many different forms. We borrow this assumption from Chen and Jiang (2019), where they used it for

non-robust offline RL. In particular, we note that the distribution $\nu$ is in the collection of discounted state-action occupancies on model $P^o$ alone for the robust RL.

**Assumption 2** (Approximate completeness). *Let $\mu \in \Delta(\mathcal{S} \times \mathcal{A})$ be the data distribution. Then, $\sup_{f \in \mathcal{F}} \inf_{f' \in \mathcal{F}} \|f' - Tf\|_{2,\mu}^2 \le \varepsilon_{\mathrm{c}}$.*

Assumption 2 states that the function class $\mathcal{F}$ is approximately closed under the robust Bellman operator $T$. This assumption has also been widely used in the offline RL literature (Agarwal et al., 2019; Chen and Jiang, 2019; Wang et al., 2021; Xie et al., 2021).

One of the most important properties that the function class $\mathcal{F}$ should have is that there must exist a function $f' \in \mathcal{F}$ which well-approximates $Q^*$. This assumption is typically called *approximate realizability* in the offline RL literature. This is typically formalized by assuming $\inf_{f \in \mathcal{F}} \|f - Tf\|_{2,\mu}^2 \le \varepsilon_{\mathrm{r}}$ (Chen and Jiang, 2019). It is known that the approximate completeness assumption and the concentratability assumption imply the realizability assumption (Chen and Jiang, 2019; Xie et al., 2021).

# 4 Robust Fitted Q-Iteration: Algorithm and Main Results

In this section, we give a step-by-step approach to overcome the challenges of the offline robust RL outlined in Section 1. We then combine these intermediate steps to obtain our proposed RFQI algorithm. We then present our main result about the performance guarantee of the RFQI algorithm, followed by a brief description about the proof approach.

## 4.1 Dual Reformulation of Robust Bellman Operator

One key challenge in directly using the standard definition of the optimal robust value function given in (2) or of the robust Bellman operator given in (3) for developing and analyzing robust RL algorithms is that both involve computing an expectation w.r.t. each model $P \in \mathcal{P}$. Given that the data is generated only according to the nominal model $P^o$, estimating these expectation values is really challenging. We show that we can overcome this difficulty through the dual reformulation of the robust Bellman operator, as given below.

**Proposition 1.** *Let $M$ be an RMDP with the uncertainty set $\mathcal{P}$ specified by (1) using the total variation distance $D(P_{s,a}, P_{s,a}^o) = (1/2)\|P_{s,a} - P_{s,a}^o\|_1$. Then, for any $Q : \mathcal{S} \times \mathcal{A} \to [0, 1/(1-\gamma)]$, the robust Bellman operator $T$ given in (3) can be equivalently written as*

$$(TQ)(s,a) = r(s,a) - \gamma \inf_{\eta \in [0, \frac{2}{\rho(1-\gamma)}]} (\mathbb{E}_{s' \sim P_{s,a}^o}[(\eta - V(s'))_+] - \eta + \rho(\eta - \inf_{s''} V(s''))_+), \quad (4)$$

*where $V(s) = \max_{a \in \mathcal{A}} Q(s,a)$. Moreover, the inner optimization problem in (4) is convex in $\eta$.*

This result mainly relies on Shapiro (2017, Section 3.2) and Duchi and Namkoong (2018, Proposition 1). Note that in (4), the expectation is now only w.r.t. the nominal model $P^o$, which opens up the possibility of using empirical estimates obtained from the data generated according to $P^o$. This avoids the need to use importance sampling based techniques to account for all models in $\mathcal{P}$, which often have high variance, and thus, are not desirable.

While (4) provides a form that is amenable to estimation using offline data, it involves finding $\inf_{s''} V(s'')$. Though this computation is straightforward in a tabular setting, it is infeasible in a function approximation setting. In order to overcome this issue, we make the following assumption.

**Assumption 3** (Fail-state). *The RMDP $M$ has a 'fail-state' $s_f$, such that $r(s_f, a) = 0$ and $P_{s_f,a}(s_f) = 1$, $\forall a \in \mathcal{A}$, $\forall P \in \mathcal{P}$.*

We note that this is not a very restrictive assumption because such a 'fail-state' is quite natural in most simulated or real-world systems. For example, a state where a robot collapses and is not able to get up, either in a simulation environment like MuJoCo or in real-world setting, is such a fail state.

Assumption 3 immediately implies that $V_{\pi,P}(s_f) = 0$, $\forall P \in \mathcal{P}$, and hence $V^*(s_f) = 0$ and $Q^*(s_f, a) = 0$, $\forall a \in \mathcal{A}$. It is also straightforward to see that $Q_{k+1}(s_f, a) = 0$, $\forall a \in \mathcal{A}$, where $Q_k$'s are the RQI iterates given by the robust Bellman update $Q_{k+1} = TQ_k$ with the initialization $Q_0 = 0$. By the contraction property of $T$, we have $Q_k \to Q^*$. So, under Assumption 3, without loss of generality, we can always keep $Q_k(s_f, a) = 0$, $\forall a \in \mathcal{A}$ and for all $k$ in RQI (and later in RFQI).

So, in the light of the above description, for the rest of the paper we will use the robust Bellman operator $T$ by setting $\inf_{s''} V(s'') = 0$. In particular, for any function $f : \mathcal{S} \times \mathcal{A} \rightarrow [0, 1/(1 - \gamma)]$ with $f(s_f, a) = 0$, the robust Bellman operator $T$ is now given by

$$(Tf)(s, a) = r(s, a) - \gamma \inf_{\eta \in [0, \frac{2}{(\rho(1-\gamma))}]} (\mathbb{E}_{s' \sim P^o_{s,a}}[(\eta - \max_{a'} f(s', a'))_+] - (1 - \rho)\eta). \quad (5)$$

## 4.2 Approximately Solving the Dual Optimization using Empirical Risk Minimization

Another key challenge in directly using the standard definition of the optimal robust value function given in (2) or of the robust Bellman operator given in (3) for developing and analyzing robust RL algorithms is that both involve an optimization over $\mathcal{P}$. The dual reformulation given in (5) partially overcomes this challenge also, as the optimization over $\mathcal{P}$ is now replaced by a convex optimization over a scalar $\eta \in [0, 2/(\rho(1 - \gamma))]$. However, this still requires solving an optimization for each $(s, a) \in \mathcal{S} \times \mathcal{A}$, which is clearly infeasible even for moderately sized state-action spaces, not to mention the function approximation setting. Our key idea to overcome this difficulty is to reformulate this as a functional optimization problem instead of solving it as multiple scalar optimization problems. This functional optimization method will make it amenable to approximately solving the dual problem using an empirical risk minimization approach with offline data.

Consider the probability (measure) space $(\mathcal{S} \times \mathcal{A}, \Sigma(\mathcal{S} \times \mathcal{A}), \mu)$ and let $L^1(\mathcal{S} \times \mathcal{A}, \Sigma(\mathcal{S} \times \mathcal{A}), \mu)$ be the set of all absolutely integrable functions defined on this space.[1] In other words, $L^1$ is the set of all functions $g : \mathcal{S} \times \mathcal{A} \rightarrow \mathcal{C} \subset \mathbb{R}$, such that $\|g\|_{1,\mu}$ is finite. We set $\mathcal{C} = [0, 2/\rho(1 - \gamma)]$, anticipating the solution of the dual optimization problem (5). We also note $\mu$ is the data generating distribution which is a $\sigma$-finite measure.

For any given function $f : \mathcal{S} \times \mathcal{A} \rightarrow [0, 1/(1 - \gamma)]$, we define the loss function $L_{\text{dual}}(\cdot; f)$ as

$$L_{\text{dual}}(g; f) = \mathbb{E}_{s,a \sim \mu}[\mathbb{E}_{s' \sim P^o_{s,a}}[(g(s, a) - \max_{a'} f(s', a'))_+] - (1 - \rho)g(s, a)], \quad \forall g \in L^1. \quad (6)$$

In the following lemma, we show that the scalar optimization over $\eta$ for each $(s, a)$ pair in (5) can be replaced by a single functional optimization w.r.t. the loss function $L_{\text{dual}}$.

**Lemma 1.** *Let $L_{\text{dual}}$ be the loss function defined in (6). Then, for any function $f : \mathcal{S} \times \mathcal{A} \rightarrow [0, 1/(1 - \gamma)]$, we have*

$$\inf_{g \in L^1} L_{\text{dual}}(g; f) = \mathbb{E}_{s,a \sim \mu}\left[ \inf_{\eta \in [0, \frac{2}{(\rho(1-\gamma))}]} \left( \mathbb{E}_{s' \sim P^o_{s,a}}[(\eta - \max_{a'} f(s', a'))_+] - (1 - \rho)\eta \right) \right]. \quad (7)$$

Note that the RHS of (7) has minimization over $\eta$ for each $(s, a)$ pair and minimization is inside the expectation $\mathbb{E}_{s,a \sim \mu}[\cdot]$. However, the LHS of (7) has a single functional minimization over $g \in L^1$ and this minimization is outside the expectation. For interchanging the expectation and minimization, and for moving from point-wise optimization to functional optimization, we use the result from Rockafellar and Wets (2009, Theorem 14.60), along with the fact that $L^1$ is a decomposable space. We also note that this result has been used in many recent works on distributionally robust optimization (Shapiro, 2017; Duchi and Namkoong, 2018) (see Appendix A for more details).

We can now define the empirical loss function $\widehat{L}_{\text{dual}}$ corresponding to the true loss $L_{\text{dual}}$ as

$$\widehat{L}_{\text{dual}}(g; f) = \frac{1}{N} \sum_{i=1}^{N} (g(s_i, a_i) - \max_{a'} f(s'_i, a'))_+ - (1 - \rho)g(s_i, a_i). \quad (8)$$

Now, for any given $f$, we can find an approximately optimal dual function through the *empirical risk minimization* approach as $\inf_{g \in L^1} \widehat{L}_{\text{dual}}(g; f)$.

As we mentioned in Section 3, our offline robust RL algorithm is given an input function class $\mathcal{G} = \{g : \mathcal{S} \times \mathcal{A} \rightarrow [0, 2/(\rho(1 - \gamma))]\}$ to approximate the dual variable functions. So, in the empirical risk minimization, instead of taking the infimum over all the functions in $L^1$, we can only take the infimum over all the functions in $\mathcal{G}$. For this to be meaningful, $\mathcal{G}$ should have sufficient representation power. In particular, the result in Lemma 1 should hold approximately even if we replace the infimum over $L^1$ with infimum over $\mathcal{G}$. One can see that this is similar to the realizability requirement for the function class $\mathcal{F}$ as described in Section 3. We formalize the representation power of $\mathcal{G}$ in the following assumption.

---

[1]In the following, we will simply denote $L^1(\mathcal{S} \times \mathcal{A}, \Sigma(\mathcal{S} \times \mathcal{A}), \mu)$ as $L^1$ for conciseness.

**Assumption 4** (Approximate dual realizability). *For all $f \in \mathcal{F}$, there exists a uniform constant $\varepsilon_{dual}$ such that $\inf_{g \in \mathcal{G}} L_{\mathrm{dual}}(g; f) - \inf_{g \in L^1} L_{\mathrm{dual}}(g; f) \leq \varepsilon_{dual}$*

Using the above assumption, for any given $f \in \mathcal{F}$, we can find an approximately optimal dual function $\widehat{g}_f \in \mathcal{G}$ through the *empirical risk minimization* approach as $\widehat{g}_f = \arg\min_{g \in \mathcal{G}} \widehat{L}_{\mathrm{dual}}(g; f)$.

In order to characterize the performance of this approach, consider the operator $T_g$ for any $g \in \mathcal{G}$ as

$$(T_g f)(s,a) = r(s,a) - \gamma(\mathbb{E}_{s' \sim P_{s,a}^o}[(g(s,a) - \max_{a'} f(s', a'))_+] - (1-\rho)g(s,a)), \quad (9)$$

for all $f \in \mathcal{F}$ and $(s,a) \in \mathcal{S} \times \mathcal{A}$. We will show in Lemma 6 in Appendix C that the error $\sup_{f \in \mathcal{F}} \|Tf - T_{\widehat{g}_f} f\|_{1,\mu}$ is $\mathcal{O}(\log(|\mathcal{F}|/\delta)/\sqrt{N})$ with probability at least $1 - \delta$.

### 4.3 Robust Fitted Q-iteration

The intuitive idea behind our robust fitted Q-iteration (RFQI) algorithm is to approximate the exact RQI update step $Q_{k+1} = TQ_k$ with function approximation using offline data. The exact RQI step requires updating each $(s,a)$-pair separately, which is not scalable to large state-action spaces. So, this is replaced by the function approximation as $Q_{k+1} = \arg\min_{f \in \mathcal{F}} \|TQ_k - f\|_{2,\mu}^2$. It is still infeasible to perform this update as it requires to exactly compute the expectation (w.r.t. $P^o$ and $\mu$) and to solve the dual problem accurately. We overcome these issues by replacing both these exact computations with empirical estimates using the offline data. We note that this intuitive idea is similar to that of the FQI algorithm in the non-robust case. However, RFQI has unique challenges due to the nature of the robust Bellman operator $T$ and the presence of the dual optimization problem within $T$.

Given a dataset $\mathcal{D}$, we also follow the standard non-robust offline RL choice of least-squares residual minimization (Chen and Jiang, 2019; Xie et al., 2021; Wang et al., 2021). Define the empirical loss of $f$ given $f'$ (which represents the $Q$-function from the last iteration) and dual variable function $g$ as

$$\widehat{L}_{\mathrm{RFQI}}(f; f', g) = \frac{1}{N} \sum_{i=1}^{N} \left( \begin{array}{c} r(s_i, a_i) + \gamma\big( -(g(s_i, a_i) - \max_{a'} f'(s_i', a'))_+ \\ + (1-\rho)g(s_i, a_i)\big) - f(s_i, a_i) \end{array} \right)^2. \quad (10)$$

The correct dual variable function to be used in (10) is the optimal dual variable $g_{f'}^* = \arg\min_{g \in \mathcal{G}} L_{\mathrm{dual}}(g; f')$ corresponding to the last iterate $f'$, which we will approximate it by $\widehat{g}_{f'} = \arg\min_{g \in \mathcal{G}} \widehat{L}_{\mathrm{dual}}(g; f')$. The RFQI update is then obtained as $\arg\min_{f \in \mathcal{F}} \widehat{L}_{\mathrm{RFQI}}(f; f', \widehat{g}_{f'})$.

Summarizing the individual steps described above, we formally give our RFQI algorithm below.

---

**Algorithm 1** Robust Fitted Q-Iteration (RFQI) Algorithm

---

1: **Input:** Offline dataset $\mathcal{D} = (s_i, a_i, r_i, s_i')_{i=1}^{N}$, function classes $\mathcal{F}$ and $\mathcal{G}$.
2: **Initialize:** $Q_0 \equiv 0 \in \mathcal{F}$.
3: **for** $k = 0, \cdots, K-1$ **do**
4:     **Dual variable function optimization:** Compute the dual variable function corresponding to $Q_k$ through empirical risk minimization as $g_k = \widehat{g}_{Q_k} = \arg\min_{g \in \mathcal{G}} \widehat{L}_{\mathrm{dual}}(g; Q_k)$ (see (8)).
5:     **Robust Q-update:** Compute the next iterate $Q_{k+1}$ through least-squares regression as $Q_{k+1} = \arg\min_{Q \in \mathcal{F}} \widehat{L}_{\mathrm{RFQI}}(Q; Q_k, g_k)$ (see (10)).
6: **end for**
7: **Output:** $\pi_K = \arg\max_a Q_K(s,a)$

---

Now we state our main theoretical result on the performance of the RFQI algorithm.

**Theorem 1.** *Let Assumptions 1-4 hold. Let $\pi_K$ be the output of the RFQI algorithm after $K$ iterations. Denote $J^\pi = \mathbb{E}_{s \sim d_0}[V^\pi(s)]$ where $d_0$ is initial state distribution. Then, for any $\delta \in (0,1)$, with probability at least $1 - 2\delta$, we have*

$$J^{\pi^*} - J^{\pi_K} \leq \frac{\gamma^K}{(1-\gamma)^2} + \frac{\sqrt{C}(\sqrt{6\varepsilon_c} + \gamma\varepsilon_{dual})}{(1-\gamma)^2} + \frac{16}{\rho(1-\gamma)^3}\sqrt{\frac{18C\log(2|\mathcal{F}||\mathcal{G}|/\delta)}{N}}.$$

*Remark* 1. Theorem 1 states that the RFQI algorithm can achieve approximate optimality. To see this, note that with $K \geq \mathcal{O}(\frac{1}{\log(1/\gamma)} \log(\frac{1}{\varepsilon(1-\gamma)}))$, and neglecting the second term corresponding to (inevitable) approximation errors $\varepsilon_c$ and $\varepsilon_{\text{dual}}$, we get $J^{\pi^*} - J^{\pi_K} \leq \varepsilon/(1-\gamma)$ with probability greater than $1 - 2\delta$ for any $\varepsilon, \delta \in (0,1)$, as long as the number of samples $N \geq \mathcal{O}(\frac{1}{(\rho\varepsilon)^2(1-\gamma)^4} \log \frac{|\mathcal{F}||\mathcal{G}|}{\delta})$. So, the above theorem can also be interpreted as a **sample complexity** result.

*Remark* 2. The known sample complexity of robust-RL in the tabular setting is $\widetilde{\mathcal{O}}(\frac{|\mathcal{S}|^2|\mathcal{A}|}{(\rho\varepsilon)^2(1-\gamma)^4})$ (Yang et al., 2021; Panaganti and Kalathil, 2022). Considering $\widetilde{\mathcal{O}}(\log(|\mathcal{F}||\mathcal{G}|))$ to be $\widetilde{\mathcal{O}}(|\mathcal{S}||\mathcal{A}|)$, we can recover the same bound as in the tabular setting (we save $|\mathcal{S}|$ due to the use of Bernstein inequality).

*Remark* 3. Under similar Bellman completeness and concentratability assumptions, RFQI sample complexity is comparable to that of a non-robust offline RL algorithm, i.e., $\mathcal{O}(\frac{1}{\varepsilon^2(1-\gamma)^4} \log \frac{|\mathcal{F}|}{\delta})$ (Chen and Jiang, 2019). As a consequence of robustness, we have $\rho^{-2}$ and $\log(|\mathcal{G}|)$ factors in our bound.

### 4.4 Proof Sketch

Here we briefly explain the key ideas used in the analysis of RFQI for obtaining the optimality gap bound in Theorem 1. The complete proof is provided in Appendix C.

*Step 1:* To bound $J^{\pi^*} - J^{\pi_K}$, we connect it to the error $\|Q^{\pi^*} - Q_K\|_{1,\nu}$ for any state-action distribution $\nu$. While the similar step follows almost immediately using the well-known performance lemma in the analysis of non-robust FQI, such a result is not known in the robust RL setting. So, we derive the basic inequalities to get a recursive form and to obtain the bound $J^{\pi^*} - J^{\pi_K} \leq 2\|Q^{\pi^*} - Q_K\|_{1,\nu}/(1-\gamma)$ (see (22) and the steps before in Appendix C).

*Step 2:* To bound $\|Q^{\pi^*} - Q_K\|_{1,\nu}$ for any state-action distribution $\nu$ such that $\|\nu/\mu\|_\infty \leq \sqrt{C}$, we decompose it to get a recursion, with approximation terms based on the least-squares regression and empirical risk minimization. Recall that $\widehat{g}_f$ is the dual variable function from the algorithm for state-action value function $f \in \mathcal{F}$. Denote $\widehat{f}_g$ as the least squares solution from the algorithm for the state-action value function $f \in \mathcal{F}$ and dual variable function $g \in \mathcal{G}$, i.e., $\widehat{f}_g = \arg\min_{Q \in \mathcal{F}} \widehat{L}_{\text{RFQI}}(Q; f, g)$. By recursive use of the obtained inequality (23) (see Appendix C) and using uniform bound, we get

$$\|Q^{\pi^*} - Q_K\|_{1,\nu} \leq \frac{\gamma^K}{1-\gamma} + \frac{\sqrt{C}}{1-\gamma} \sup_{f \in \mathcal{F}} \|Tf - T_{\widehat{g}_f}f\|_{1,\mu} + \frac{\sqrt{C}}{1-\gamma} \sup_{f \in \mathcal{F}} \sup_{g \in \mathcal{G}} \|T_g f - \widehat{f}_g\|_{2,\mu}.$$

*Step 3:* We recognize that $\sup_{f \in \mathcal{F}} \|Tf - T_{\widehat{g}_f}f\|_{1,\mu}$ is an empirical risk minimization error term. Using Rademacher complexity based bounds, we show in Lemma 6 that this error is $\mathcal{O}(\log(|\mathcal{F}|/\delta)/\sqrt{N})$ with high probability.

*Step 4:* Similarly, we also recognize that $\sup_{f \in \mathcal{F}} \sup_{g \in \mathcal{G}} \|T_g f - \widehat{f}_g\|_{2,\mu}$ is a least-squares regression error term. We also show that this error is $\mathcal{O}(\log(|\mathcal{F}||\mathcal{G}|/\delta)/\sqrt{N})$ with high probability. We adapt the generalized least squares regression result to accommodate the modified target functions resulting from the robust Bellman operator to obtain this bound (see Lemma 7).

The proof is complete after combining steps 1-4 above.

## 5  Experiments

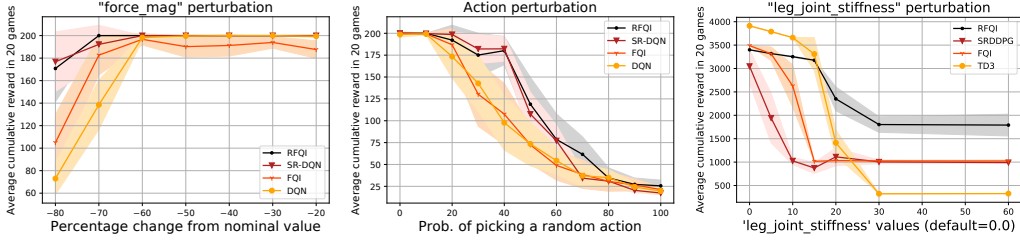

Figure 1: CartPole     Figure 2: CartPole     Figure 3: Hopper

Here, we demonstrate the robust performance of our RFQI algorithm by evaluating it on *Cartpole* and *Hopper* environments in OpenAI Gym (Brockman et al., 2016). In all the figures shown, the quantity

in the vertical axis is averaged over 20 different seeded runs depicted by the thick line and the band around it is the $\pm 0.5$ standard deviation. *A more detailed description of the experiments, and results on additional experiments, are deferred to Appendix E.* We provide our code in **github webpage** `https://github.com/zaiyan-x/RFQI` containing instructions to reproduce all results in this paper.

For the *Cartpole*, we compare RFQI algorithm against the non-robust RL algorithms FQI and DQN, and the soft-robust RL algorithm proposed in Derman et al. (2018). We test the robustness of the algorithms by changing the parameter *force_mag* (to model external force disturbance), and also by introducing action perturbations (to model actuator noise). Fig. 1 and Fig. 2 shows superior robust performance of RFQI compared to the non-robust FQI and DQN. The RFQI performance is similar to that of soft-robust DQN. We note that soft-robust RL algorithm (here soft-robust DQN) is an online deep RL algorithm (and not an offline RL algorithm) and has no provable performance guarantee. Moreover, soft-robust RL algorithm requires generating online data according a number of models in the uncertainty set, whereas RFQI only requires offline data according to a single nominal training model.

For the *Hopper*, we compare RFQI algorithm against the non-robust RL algorithms FQI and TD3 (Fujimoto et al., 2018), and the soft-robust RL (here soft-robust DDPG) algorithm proposed in Derman et al. (2018). We test the robustness of the algorithms by changing the parameter *leg_joint_stiffness*. Fig. 3 shows the superior performance of our RFQI algorithm against the non-robust algorithms and soft-robust DDPG algorithm. The average episodic reward of RFQI remains almost the same initially, and later decays much less and gracefully when compared to the non-robust FQI and TD3.

## 6  Conclusion

In this work, we presented a novel robust RL algorithm called Robust Fitted Q-Iteration algorithm with provably optimal performance for an RMDP with arbitrarily large state space, using only offline data with function approximation. We also demonstrated the superior performance of the proposed algorithm on standard benchmark problems.

One limitation of our present work is that, we considered only the uncertainty set defined with respect to the total variation distance. In future work, we will consider uncertainty sets defined with respect to other $f$-divergences such as KL-divergence and Chi-square divergence. Finding a lower bound for the sample complexity and relaxing the assumptions used are also important and challenging problems.

## 7  Acknowledgements

This work was supported in part by the National Science Foundation (NSF) grants NSF-CAREER-EPCN-2045783 and NSF ECCS 2038963. Any opinions, findings, and conclusions or recommendations expressed in this material are those of the authors and do not necessarily reflect the views of the sponsoring agencies.

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
