# Appendix

## A  Useful Technical Results

In this section, we state some existing results from concentration inequalities, generalization bounds, and optimization theory that we will use later in our analysis. We first state the Berstein's inequality that utilizes second-moment to get a tighter concentration inequality.

**Lemma 2** (Bernstein's inequality (Vershynin, 2018, Theorem 2.8.4))**.** *Let $X_1, \cdots, X_T$ be independent random variables. Assume that $|X_t - \mathbb{E}[X_t]| \leq M$, for all $t$. Then, for any $\varepsilon > 0$, we have*

$$\mathbb{P}\left(\left|\frac{1}{T}\sum_{t=1}^{T}(X_t - \mathbb{E}[X_t])\right| \geq \varepsilon\right) \leq 2\exp\left(-\frac{T^2\varepsilon^2}{2\sigma^2 + \frac{2MT\varepsilon}{3}}\right),$$

*where $\sigma^2 = \sum_{t=1}^{T}\mathbb{E}[X_t^2]$. Furthermore, if $X_1, \cdots, X_T$ are independent and identically distributed random variables, then for any $\delta \in (0, 1)$, we have*

$$\left|\mathbb{E}[X_1] - \frac{1}{T}\sum_{t=1}^{T}X_t\right| \leq \sqrt{\frac{2\mathbb{E}[X_1^2]\log(2/\delta)}{T}} + \frac{M\log(2/\delta)}{3T},$$

*with probability at least $1 - \delta$.*

We now state a result for the generalization bounds on empirical risk minimization (ERM) problems. This result is adapted from Shalev-Shwartz and Ben-David (2014, Theorem 26.5, Lemma 26.8, Lemma 26.9).

**Lemma 3** (ERM generalization bound)**.** *Let $P$ be the data generating distribution on the space $\mathcal{X}$ and let $\mathcal{H}$ be a given hypothesis class of functions. Assume that for all $x \in \mathcal{X}$ and $h \in \mathcal{H}$ we have that $|l(h, x)| \leq c_1$ for some positive constant $c_1 > 0$. Given a dataset $\mathcal{D} = \{X_i\}_{i=1}^{N}$, generated independently from $P$, denote $\hat{h}$ as the ERM solution, i.e. $\hat{h} = \arg\min_{h \in \mathcal{H}}(1/N)\sum_{i=1}^{N}l(h, X_i)$. For any fixed $\delta \in (0, 1)$ and $h^* \in \arg\min_{h \in \mathcal{H}}\mathbb{E}_{X \sim P}[l(h, X)]$, we have*

$$\mathbb{E}_{X \sim P}[l(\hat{h}, X)] - \mathbb{E}_{X \sim P}[l(h^*, X)] \leq 2R(l \circ \mathcal{H} \circ \mathcal{D}) + 5c_1\sqrt{\frac{2\log(8/\delta)}{N}}, \tag{11}$$

*with probability at least $1 - \delta$, where $R(\cdot)$ is the Rademacher complexity of $l \circ \mathcal{H}$ given by*

$$R(l \circ \mathcal{H} \circ \mathcal{D}) = \frac{1}{N}\mathbb{E}_{\{\sigma_i\}_{i=1}^{N}}\left(\sup_{g \in l \circ \mathcal{H}}\sum_{i=1}^{N}\sigma_i g(X_i)\right),$$

*in which $\sigma_i$'s are independent from $X_i$'s and are independently and identically distributed according to the Rademacher random variable $\sigma$, i.e. $\mathbb{P}(\sigma = 1) = 0.5 = \mathbb{P}(\sigma = -1)$.*

*Furthermore, if $\mathcal{H}$ is a finite hypothesis class, i.e. $|\mathcal{H}| < \infty$, with $|h \circ x| \leq c_2$ for all $h \in \mathcal{H}$ and $x \in \mathcal{X}$, and $l(h, x)$ is $c_3$-Lipschitz in $h$, then we have*

$$\mathbb{E}_{X \sim P}[l(\hat{h}, X)] - \mathbb{E}_{X \sim P}[l(h^*, X)] \leq 2c_2 c_3\sqrt{\frac{2\log(|\mathcal{H}|)}{N}} + 5c_1\sqrt{\frac{2\log(8/\delta)}{N}}, \tag{12}$$

*with probability at least $1 - \delta$.*

We now mention two important concepts from variational analysis (Rockafellar and Wets, 2009) literature that is useful to relate minimization of integrals and the integrals of pointwise minimization under special class of functions.

**Definition 1** (Decomposable spaces and Normal integrands (Rockafellar and Wets, 2009, Definition 14.59, Example 14.29))**.** *A space $\mathcal{X}$ of measurable functions is a decomposable space relative to an underlying measure space $(\Omega, \mathcal{A}, \mu)$, if for every function $x_0 \in \mathcal{X}$, every set $A \in \mathcal{A}$ with $\mu(A) < \infty$, and any bounded measurable function $x_1 : A \to \mathbb{R}$, the function $x(\omega) = x_0(\omega)\mathbb{1}(\omega \notin A) + x_1(\omega)\mathbb{1}(\omega \in A)$ belongs to $\mathcal{X}$. A function $f : \Omega \times \mathbb{R} \to \mathbb{R}$ (finite-valued) is a normal integrand, if and only if $f(\omega, x)$ is $\mathcal{A}$-measurable in $\omega$ for each $x$ and is continuous in $x$ for each $\omega$.*

*Remark* 4. A few examples of decomposable spaces are $L^p(\mathcal{S} \times \mathcal{A}, \Sigma(\mathcal{S} \times \mathcal{A}), \mu)$ for any $p \geq 1$ and $\mathcal{M}(\mathcal{S} \times \mathcal{A}, \Sigma(\mathcal{S} \times \mathcal{A}))$, the space of all $\Sigma(\mathcal{S} \times \mathcal{A})$-measurable functions.

**Lemma 4** (Rockafellar and Wets, 2009, Theorem 14.60). *Let $\mathcal{X}$ be a space of measurable functions from $\Omega$ to $\mathbb{R}$ that is decomposable relative to a $\sigma$-finite measure $\mu$ on the $\sigma$-algebra $\mathcal{A}$. Let $f : \Omega \times \mathbb{R} \to \mathbb{R}$ (finite-valued) be a normal integrand. Then, we have*

$$\inf_{x \in \mathcal{X}} \int_{\omega \in \Omega} f(\omega, x(\omega))\mu(\mathrm{d}\,\omega) = \int_{\omega \in \Omega} \left( \inf_{x \in \mathbb{R}} f(\omega, x) \right) \mu(\mathrm{d}\,\omega).$$

*Moreover, as long as the above infimum is not $-\infty$, we have that*

$$x' \in \arg\min_{x \in \mathcal{X}} \int_{\omega \in \Omega} f(\omega, x(\omega))\mu(\mathrm{d}\,\omega),$$

*if and only if $x'(\omega) \in \arg\min_{x \in \mathbb{R}} f(\omega, x) \cdot \mu$ almost surely.*

We now give one result from distributioanlly robust optimization. The $f$-divergence between the distributions $P$ and $P^o$ is defined as

$$D_f(P\|P^o) = \int f(\frac{\mathrm{d}\,P}{\mathrm{d}\,P^o})\mathrm{d}\,P^o, \tag{13}$$

where $f$ is a convex function (Csiszár, 1967; Moses and Sundaresan, 2011). We obtain different divergences for different forms of the function $f$, including some well-known divergences. For example, $f(t) = |t - 1|/2$ gives Total Variation (TV), $f(t) = t \log t$ gives Kullback-Leibler (KL), $f(t) = (t - 1)^2$ gives Chi-square, and $f(t) = (\sqrt{t} - 1)^2$ gives squared Hellinger divergences.

Let $P^o$ be a distribution on the space $\mathcal{X}$ and let $l : \mathcal{X} \to \mathbb{R}$ be a loss function. We have the following result from the *distributionally robust optimization* literature, see e.g., Shapiro (2017, Section 3.2) and Duchi and Namkoong (2018, Proposition 1).

**Proposition 2.** *Let $D_f$ be the $f$-divergence as defined in* (13). *Then,*

$$\sup_{D_f(P\|P^o)\leq\rho} \mathbb{E}_P[l(X)] = \inf_{\lambda>0,\eta\in\mathbb{R}} \mathbb{E}_{P^o}\left[\lambda f^*\left(\frac{l(X)-\eta}{\lambda}\right)\right] + \lambda\rho + \eta, \tag{14}$$

*where $f^*(s) = \sup_{t\geq 0}\{st - f(t)\}$ is the Fenchel conjugate.*

Note that on the right hand side of (14), the expectation is taken only with respect to $P^o$. We will use the above result to derive the dual reformulation of the robust Bellman operator.

# B   Proof of the Proposition 1

As the first step, we adapt the result given in Proposition 2 in two ways: $(i)$ Since Proposition 1 considers the TV uncertainty set, we will derive the specific form of this result for the TV uncertainty set, $(ii)$ Since Proposition 1 considers the minimization problem instead of the maximization problem, unlike in Proposition 2, we will derive the specific form of this result for minimization.

**Lemma 5.** *Let $D_f$ be as defined in* (13) *with $f(t) = |t - 1|/2$ corresponding to the TV uncertainty set. Then,*

$$\inf_{D_f(P\|P^o)\leq\rho} \mathbb{E}_P[l(X)] = -\inf_{\eta\in\mathbb{R}} \mathbb{E}_{P^o}[(\eta - l(X))_+] + (\eta - \inf_{x\in\mathcal{X}} l(x))_+ \times \rho - \eta,$$

*Proof.* First, we will compute the Fenchel conjugate of $f(t) = |t - 1|/2$. We have

$$f^*(s) = \sup_{t\geq 0} \{st - \frac{1}{2}|t - 1|\} = \max\{\sup_{t\in[0,1]} \{(s + \frac{1}{2})t - \frac{1}{2}\}, \sup_{t>1}\{(s - \frac{1}{2})t + \frac{1}{2}\}\}.$$

It is easy to see that for $s > 1/2$, we have $f^*(s) = +\infty$, and for $s \leq -1/2$, we have $f^*(s) = -1/2$. For $s \in [-1/2, 1/2]$, we have

$$f^*(s) = \max\{\sup_{t\in[0,1]} \{(s + \frac{1}{2})t - \frac{1}{2}\}, \sup_{t>1}(\{(s - \frac{1}{2})t + \frac{1}{2}\}\}$$

$$= \max\left\{\left(\left(s + \frac{1}{2}\right) \cdot 1 - \frac{1}{2}\right), \left(\left(s - \frac{1}{2}\right) \cdot 1 + \frac{1}{2}\right)\right\} = s.$$

Thus, we have

$$f^*(s) = \begin{cases} -\frac{1}{2} & s \leq -\frac{1}{2}, \\ s & s \in [-\frac{1}{2}, \frac{1}{2}]. \\ +\infty & s > \frac{1}{2}. \end{cases}$$

From Proposition 2, we obtain

$$\sup_{D_f(P\|P^o)\leq\rho} \mathbb{E}_P[l(X)] = \inf_{\lambda>0,\eta\in\mathbb{R}} \mathbb{E}_{P^o}\left[\lambda f^*\left(\frac{l(X)-\eta}{\lambda}\right)\right] + \lambda\rho + \eta$$

$$= \inf_{\lambda,\eta:\lambda>0,\eta\in\mathbb{R},\frac{\sup_{x\in\mathcal{X}} l(x)-\eta}{\lambda}\leq\frac{1}{2}} \mathbb{E}_{P^o}\left[\lambda \max\left\{\frac{l(X)-\eta}{\lambda}, -\frac{1}{2}\right\}\right] + \lambda\rho + \eta$$

$$= \inf_{\lambda,\eta:\lambda>0,\eta\in\mathbb{R},\frac{\sup_{x\in\mathcal{X}} l(x)-\eta}{\lambda}\leq\frac{1}{2}} \mathbb{E}_{P^o}[\max\{l(X)-\eta, -\lambda/2\}] + \lambda\rho + \eta$$

$$= \inf_{\lambda,\eta:\lambda>0,\eta\in\mathbb{R},\frac{\sup_{x\in\mathcal{X}} l(x)-\eta}{\lambda}\leq\frac{1}{2}} \mathbb{E}_{P^o}[(l(X)-\eta+\lambda/2)_+] - \lambda/2 + \lambda\rho + \eta$$

$$= \inf_{\lambda,\eta':\lambda>0,\eta'\in\mathbb{R},\frac{\sup_{x\in\mathcal{X}} l(x)-\eta'}{\lambda}\leq 1} \mathbb{E}_{P^o}[(l(X)-\eta')_+] + \lambda\rho + \eta'.$$

The second equality follows since $f^*\left(\frac{l(X)-\eta}{\lambda}\right) = +\infty$ whenever $\frac{l(X)-\eta}{\lambda} > \frac{1}{2}$, which can be ignored as we are minimizing over $\lambda$ and $\eta$. The fourth equality follows form the fact that $\max\{x,y\} = (x-y)_+ + y$ for any $x, y \in \mathbb{R}$. Finally, the last equality follows by making the substitution $\eta' = \eta - \lambda/2$. Taking the optimal value of $\lambda$, i.e., $\lambda = (\sup_{x\in\mathcal{X}} l(x) - \eta')_+$, we get

$$\sup_{D_f(P\|P^o)\leq\rho} \mathbb{E}_P[l(X)] = \inf_{\eta\in\mathbb{R}} \mathbb{E}_{P^o}[(l(X)-\eta)_+] + (\sup_{x\in\mathcal{X}} l(x) - \eta)_+\rho + \eta.$$

Now,

$$\inf_{D_f(P\|P^o)\leq\rho} \mathbb{E}_P[l(X)] = -\sup_{D_f(P\|P^o)\leq\rho} \mathbb{E}_P[-l(X)]$$

$$= -\inf_{\eta\in\mathbb{R}} \mathbb{E}_{P^o}[(-l(X)-\eta)_+] + (\sup_{x\in\mathcal{X}} -l(x) - \eta)_+\rho + \eta$$

$$= -\inf_{\eta'\in\mathbb{R}} \mathbb{E}_{P^o}[(\eta'-l(X))_+] + (\eta' - \inf_{x\in\mathcal{X}} l(x))_+\rho - \eta',$$

which completes the proof. $\qquad\square$

We are now ready to prove Proposition 1.

***Proof of Proposition 1.*** For each $(s,a)$, the optimization problem in (3) is given by $\min_{P_{s,a}\in\mathcal{P}_{s,a}} \mathbb{E}_{s'\sim P_{s,a}}[V(s')]$, and our focus is on the setting where $\mathcal{P}_{s,a}$ is given by the TV uncertainty set. So, $\mathcal{P}_{s,a}$ can be equivalently defined using the $f$-divergence with $f(t) = |t-1|/2$ as $\mathcal{P}_{s,a} = \{P_{s,a} : D_f(P_{s,a}\|P_{s,a}^o) \leq \rho\}$. We can now use the result of Lemma 5 to get

$$\inf_{P_{s,a}\in\mathcal{P}_{s,a}} \mathbb{E}_{s'\sim P_{s,a}}[V(s')] = -\inf_{\eta\in\mathbb{R}} \mathbb{E}_{s'\sim P_{s,a}^o}[(\eta-V(s'))_+] + (\eta - \inf_{s''\in\mathcal{S}} V(s''))_+\rho - \eta.$$

From Proposition 2, the function $h(\eta) = \mathbb{E}_{s'\sim P_{s,a}^o}[(\eta-V(s'))_+] + \rho(\eta - \inf_{s''} V(s''))_+ - \eta$ is convex in $\eta$. Since $V(s') \geq 0$, $h(\eta) = -\eta \geq 0$ when $\eta \leq 0$. So, $\inf_{\eta\in(-\infty,0]} h(\eta)$, achieved at $\eta = 0$. Also, since $V(s) \leq 1/(1-\gamma)$, we have

$$h\left(\frac{2}{\rho(1-\gamma)}\right) = \mathbb{E}_{s'\sim P_{s,a}^o}\left[\frac{2}{\rho(1-\gamma)} - V(s')\right] + \rho\left(\frac{2}{\rho(1-\gamma)} - \inf_{s''} V(s'')\right) - \frac{2}{\rho(1-\gamma)}$$

$$\geq -\frac{1}{(1-\gamma)} + \rho\left(\frac{2}{\rho(1-\gamma)} - \frac{1}{(1-\gamma)}\right) = \frac{2}{(1-\gamma)} - \frac{(1+\rho)}{(1-\gamma)} \geq 0.$$

So, it is sufficient to consider $\eta \in [0, \frac{2}{\rho(1-\gamma)}]$ for the above optimization problem.

Using these, we get

$$(TQ)(s,a) = r(s,a) + \gamma \inf_{P_{s,a} \in \mathcal{P}_{s,a}} \mathbb{E}_{s' \sim P_{s,a}}[V(s')]$$

$$= r(s,a) + \gamma \cdot -1 \cdot \inf_{\eta \in [0, \frac{2}{\rho(1-\gamma)}]} \mathbb{E}_{s' \sim P_{s,a}^o}[(\eta - V(s'))_+] + (\eta - \inf_{s'' \in \mathcal{S}} V(s''))_+ \rho - \eta.$$

This completes the proof of Proposition 1. $\qquad\square$

## C  Proof of Theorem 1

We start by proving Lemma 1 which mainly follows from Lemma 4 in Appendix A.

*Proof of Lemma 1.* Let $h((s,a),\eta) = \mathbb{E}_{s' \sim P_{s,a}^o}((\eta - \max_{a'} f(s',a'))_+ - (1-\rho)\eta)$. We note that $h((s,a),\eta)$ is $\Sigma(\mathcal{S} \times \mathcal{A})$-measurable in $(s,a) \in \mathcal{S} \times \mathcal{A}$ for each $\eta \in [0, 1/(\rho(1-\gamma))]$ and is continuous in $\eta$ for each $(s,a) \in \mathcal{S} \times \mathcal{A}$. Now it follows that $h((s,a),\eta)$ is a normal integrand (see Definition 1 in Appendix A). We now note that $L^1(\mathcal{S} \times \mathcal{A}, \Sigma(\mathcal{S} \times \mathcal{A}), \mu)$ is a decomposable space (Remark 4 in Appendix A). Thus, this lemma now directly follows from Lemma 4. $\qquad\square$

Now we state a result and provide its proof for the empirical risk minimization on the dual parameter.

**Lemma 6** (Dual Optimization Error Bound). *Let $\widehat{g}_f$ be the dual optimization parameter from the algorithm (Step 4) for the state-action value function $f$ and let $T_g$ be as defined in* (9). *With probability at least $1 - \delta$, we have*

$$\sup_{f \in \mathcal{F}} \|Tf - T_{\widehat{g}_f} f\|_{1,\mu} \le \frac{4\gamma(2-\rho)}{\rho(1-\gamma)} \sqrt{\frac{2 \log(|\mathcal{G}|)}{N}} + \frac{25\gamma}{\rho(1-\gamma)} \sqrt{\frac{2 \log(8|\mathcal{F}|/\delta)}{N}} + \gamma\varepsilon_{dual}.$$

*Proof.* Fix an $f \in \mathcal{F}$. We will also invoke union bound for the supremum here. We recall from (8) that $\widehat{g}_f = \arg\min_{g \in \mathcal{G}} \widehat{L}_{\mathrm{dual}}(g; f)$. From the robust Bellman equation, we directly obtain

$$\|T_{\widehat{g}_f} f - Tf\|_{1,\mu} = \gamma(\mathbb{E}_{s,a \sim \mu}|\mathbb{E}_{s' \sim P_{s,a}^o}((\widehat{g}_f(s,a) - \max_{a'} f(s',a'))_+ - (1-\rho)\widehat{g}_f(s,a))$$

$$- \inf_{\eta \in [0, 2/(\rho(1-\gamma))]} \mathbb{E}_{s' \sim P_{s,a}^o}((\eta - \max_{a'} f(s',a'))_+ - (1-\rho)\eta)|)$$

$$\overset{(a)}{=} \gamma(\mathbb{E}_{s,a \sim \mu}\mathbb{E}_{s' \sim P_{s,a}^o}((\widehat{g}_f(s,a) - \max_{a'} f(s',a'))_+ - (1-\rho)\widehat{g}_f(s,a))$$

$$- \mathbb{E}_{s,a \sim \mu}[\inf_{\eta \in [0, 2/(\rho(1-\gamma))]} \mathbb{E}_{s' \sim P_{s,a}^o}((\eta - \max_{a'} f(s',a'))_+ - (1-\rho)\eta)])$$

$$\overset{(b)}{=} \gamma(\mathbb{E}_{s,a \sim \mu, s' \sim P_{s,a}^o}((\widehat{g}_f(s,a) - \max_{a'} f(s',a'))_+ - (1-\rho)\widehat{g}_f(s,a))$$

$$- \inf_{g \in L^1} \mathbb{E}_{s,a \sim \mu, s' \sim P_{s,a}^o}((g(s,a) - \max_{a'} f(s',a'))_+ - (1-\rho)g(s,a)))$$

$$= \gamma(\mathbb{E}_{s,a \sim \mu, s' \sim P_{s,a}^o}((\widehat{g}_f(s,a) - \max_{a'} f(s',a'))_+ - (1-\rho)\widehat{g}_f(s,a))$$

$$- \inf_{g \in \mathcal{G}} \mathbb{E}_{s,a \sim \mu, s' \sim P_{s,a}^o}((g(s,a) - \max_{a'} f(s',a'))_+ - (1-\rho)g(s,a)))$$

$$+ \gamma(\inf_{g \in \mathcal{G}} \mathbb{E}_{s,a \sim \mu, s' \sim P_{s,a}^o}((g(s,a) - \max_{a'} f(s',a'))_+ - (1-\rho)g(s,a))$$

$$- \inf_{g \in L^1} \mathbb{E}_{s,a \sim \mu, s' \sim P_{s,a}^o}((g(s,a) - \max_{a'} f(s',a'))_+ - (1-\rho)g(s,a)))$$

$$\overset{(c)}{\le} \gamma(\mathbb{E}_{s,a \sim \mu, s' \sim P_{s,a}^o}((\widehat{g}_f(s,a) - \max_{a'} f(s',a'))_+ - (1-\rho)\widehat{g}_f(s,a))$$

$$- \inf_{g \in \mathcal{G}} \mathbb{E}_{s,a \sim \mu, s' \sim P_{s,a}^o}((g(s,a) - \max_{a'} f(s',a'))_+ - (1-\rho)g(s,a))) + \gamma\varepsilon_{\mathrm{dual}}$$

$$\overset{(d)}{\le} 2\gamma R(l \circ \mathcal{G} \circ \mathcal{D}) + \frac{25\gamma}{\rho(1-\gamma)} \sqrt{\frac{2 \log(8/\delta)}{N}} + \gamma\varepsilon_{\mathrm{dual}}$$

$$\overset{(e)}{\le} \frac{4\gamma(2-\rho)}{\rho(1-\gamma)} \sqrt{\frac{2 \log(|\mathcal{G}|)}{N}} + \frac{25\gamma}{\rho(1-\gamma)} \sqrt{\frac{2 \log(8/\delta)}{N}} + \gamma\varepsilon_{\mathrm{dual}}.$$

$(a)$ follows since $\inf_g h(g) \leq h(\widehat{g}_f)$. $(b)$ follows from Lemma 1. $(c)$ follows from the approximate dual realizability assumption (Assumption 4).

For $(d)$, we consider the loss function $l(g, (s, a, s')) = (g(s, a) - \max_{a'} f(s', a'))_+ - (1 - \rho)g(s, a)$ and dataset $\mathcal{D} = \{s_i, a_i, s_i'\}_{i=1}^N$. Note that $|l(g, (s, a, s'))| \leq 5/(\rho(1 - \gamma))$ (since $f \in \mathcal{F}$ and $g \in \mathcal{G}$). Now, we can apply the empirical risk minimization result (11) in Lemma 3 to get $(d)$, where $R(\cdot)$ is the Rademacher complexity.

Finally, $(e)$ follows from (12) in Lemma 3 when combined with the facts that $l(g, (s, a, s'))$ is $(2 - \rho)$-Lipschitz in $g$ and $g(s, a) \leq 2/(\rho(1 - \gamma))$, since $g \in \mathcal{G}$.

With union bound, with probability at least $1 - \delta$, we finally get

$$\sup_{f \in \mathcal{F}} \|Tf - T_{\widehat{g}_f} f\|_{1,\mu} \leq \frac{4\gamma(2 - \rho)}{\rho(1 - \gamma)} \sqrt{\frac{2\log(|\mathcal{G}|)}{N}} + \frac{25\gamma}{\rho(1 - \gamma)} \sqrt{\frac{2\log(8|\mathcal{F}|/\delta)}{N}} + \gamma\varepsilon_{\text{dual}},$$

which concludes the proof. $\qquad\square$

We next prove the least-squares generalization bound for the RFQI algorithm.

**Lemma 7** (Least squares generalization bound). *Let $\widehat{f}_g$ be the least-squares solution from the algorithm (Step 5) for the state-action value function $f$ and dual variable function $g$. Let $T_g$ be as defined in* (9). *Then, with probability at least $1 - \delta$, we have*

$$\sup_{f \in \mathcal{F}} \sup_{g \in \mathcal{G}} \|T_g f - \widehat{f}_g\|_{2,\mu} \leq \sqrt{6\varepsilon_c} + \frac{16}{\rho(1 - \gamma)} \sqrt{\frac{18\log(2|\mathcal{F}||\mathcal{G}|/\delta)}{N}}.$$

*Proof.* We adapt the least-squares generalization bound given in Agarwal et al. (2019, Lemma A.11) to our setting. We recall from (10) that $\widehat{f}_g = \arg\min_{Q \in \mathcal{F}} \widehat{L}_{\text{RFQI}}(Q; f, g)$. We first fix functions $f \in \mathcal{F}$ and $g \in \mathcal{G}$. For any function $f' \in \mathcal{F}$, we define random variables $z_i^{f'}$ as

$$z_i^{f'} = (f'(s_i, a_i) - y_i)^2 - ((T_g f)(s_i, a_i) - y_i)^2,$$

where $y_i = r_i - \gamma(g(s_i, a_i) - \max_{a'} f(s_i', a'))_+ + \gamma(1 - \rho)g(s_i, a_i)$, and $(s_i, a_i, s_i') \in \mathcal{D}$ with $(s_i, a_i) \sim \mu, s_i' \sim P_{s_i, a_i}^o$. It is straightforward to note that for a given $(s_i, a_i)$, we have $\mathbb{E}_{s_i' \sim P_{s_i, a_i}^o}[y_i] = (T_g f)(s_i, a_i)$.

Also, since $g(s_i, a_i) \leq 2/(\rho(1 - \gamma))$ (because $g \in \mathcal{G}$) and $f(s_i, a_i), f'(s_i, a_i) \leq 1/(1 - \gamma)$ (because $f, f' \in \mathcal{F}$), we have $(T_g f)(s_i, a_i) \leq 5/(\rho(1 - \gamma))$. This also gives us that $y_i \leq 5/(\rho(1 - \gamma))$.

Using this, we obtain the first moment and an upper-bound for the second moment of $z_i^{f'}$ as follows:

$$\mathbb{E}_{s_i' \sim P_{s_i, a_i}^o}[z_i^{f'}] = \mathbb{E}_{s_i' \sim P_{s_i, a_i}^o}[(f'(s_i, a_i) - (T_g f)(s_i, a_i)) \cdot (f'(s_i, a_i) + (T_g f)(s_i, a_i) - 2y_i)]$$
$$= (f'(s_i, a_i) - (T_g f)(s_i, a_i))^2,$$
$$\mathbb{E}_{s_i' \sim P_{s_i, a_i}^o}[(z_i^{f'})^2] = \mathbb{E}_{s_i' \sim P_{s_i, a_i}^o}[(f'(s_i, a_i) - (T_g f)(s_i, a_i))^2 \cdot (f'(s_i, a_i) + (T_g f)(s_i, a_i) - 2y_i)^2]$$
$$= (f'(s_i, a_i) - (T_g f)(s_i, a_i))^2 \cdot \mathbb{E}_{s_i' \sim P_{s_i, a_i}^o}[(f'(s_i, a_i) + (T_g f)(s_i, a_i) - 2y_i)^2]$$
$$\leq C_1 (f'(s_i, a_i) - (T_g f)(s_i, a_i))^2,$$

where $C_1 = 16^2/(\rho^2(1 - \gamma)^2)$. This immediately implies that

$$\mathbb{E}_{s_i, a_i \sim \mu, s_i' \sim P_{s_i, a_i}^o}[z_i^{f'}] = \|T_g f - f'\|_{2,\mu}^2,$$
$$\mathbb{E}_{s_i, a_i \sim \mu, s_i' \sim P_{s_i, a_i}^o}[(z_i^{f'})^2] \leq C_1 \|T_g f - f'\|_{2,\mu}^2.$$

From these calculations, it is also straightforward to see that $|z_i^{f'} - \mathbb{E}_{s_i, a_i \sim \mu, s_i' \sim P_{s_i, a_i}^o}[z_i^{f'}]| \leq 2C_1$ almost surely.

Now, using the Bernstein's inequality (Lemma 2), together with a union bound over all $f' \in \mathcal{F}$, with probability at least $1 - \delta$, we have

$$\left| \|T_g f - f'\|_{2,\mu}^2 - \frac{1}{N} \sum_{i=1}^N z_i^{f'} \right| \leq \sqrt{\frac{2C_1 \|T_g f - f'\|_{2,\mu}^2 \log(2|\mathcal{F}|/\delta)}{N}} + \frac{2C_1 \log(2|\mathcal{F}|/\delta)}{3N}, \quad (15)$$

for all $f' \in \mathcal{F}$. Setting $f' = \widehat{f}_g$, with probability at least $1 - \delta/2$, we have

$$\|T_g f - \widehat{f}_g\|_{2,\mu}^2 \leq \frac{1}{N} \sum_{i=1}^N z_i^{\widehat{f}_g} + \sqrt{\frac{2C_1 \|T_g f - \widehat{f}_g\|_{2,\mu}^2 \log(4|\mathcal{F}|/\delta)}{N}} + \frac{2C_1 \log(4|\mathcal{F}|/\delta)}{3N}. \quad (16)$$

Now we upper-bound $(1/N) \sum_{i=1}^N z_i^{\widehat{f}_g}$ in the following. Consider a function $\widetilde{f} \in \arg\min_{h \in \mathcal{F}} \|h - T_g f\|_{2,\mu}^2$. Note that $\widetilde{f}$ is independent of the dataset. We note that our earlier first and second moment calculations hold true for $\widetilde{f}$, replacing $f'$, as well. Now, from (15) setting $f' = \widetilde{f}$, with probability at least $1 - \delta/2$ we have

$$\frac{1}{N} \sum_{i=1}^N z_i^{\widetilde{f}} - \|T_g f - \widetilde{f}\|_{2,\mu}^2 \leq \sqrt{\frac{2C_1 \|T_g f - \widetilde{f}\|_{2,\mu}^2 \log(4|\mathcal{F}|/\delta)}{N}} + \frac{2C_1 \log(4|\mathcal{F}|/\delta)}{3N}. \quad (17)$$

Suppose $(1/N) \sum_{i=1}^N z_i^{\widetilde{f}} \geq 2C_1 \log(4|\mathcal{F}|/\delta)/N$ holds, then from (17) we get

$$\frac{1}{N} \sum_{i=1}^N z_i^{\widetilde{f}} - \|T_g f - \widetilde{f}\|_{2,\mu}^2 \leq \sqrt{\|T_g f - \widetilde{f}\|_{2,\mu}^2 \cdot \frac{1}{N} \sum_{i=1}^N z_i^{\widetilde{f}}} + \frac{2C_1 \log(4|\mathcal{F}|/\delta)}{N}. \quad (18)$$

We note the following algebra fact: Suppose $x^2 - ax + b \leq 0$ with $b > 0$ and $a^2 \geq 4b$, then we have $x \leq a$. Taking $x = (1/N) \sum_{i=1}^N z_i^{\widetilde{f}}$ in this fact, from (18) we get

$$\frac{1}{N} \sum_{i=1}^N z_i^{\widetilde{f}} \leq 3\|T_g f - \widetilde{f}\|_{2,\mu}^2 + \frac{4C_1 \log(4|\mathcal{F}|/\delta)}{3N} \leq 3\|T_g f - \widetilde{f}\|_{2,\mu}^2 + \frac{2C_1 \log(4|\mathcal{F}|/\delta)}{N}. \quad (19)$$

Now suppose $(1/N) \sum_{i=1}^N z_i^{\widetilde{f}} \leq 2C_1 \log(4|\mathcal{F}|/\delta)/N$, then (19) holds immediately. Thus, (19) always holds with probability at least $1 - \delta/2$. Furthermore, recall $\widetilde{f} \in \arg\min_{h \in \mathcal{F}} \|h - T_g f\|_{2,\mu}^2$, we have

$$\frac{1}{N} \sum_{i=1}^N z_i^{\widetilde{f}} \leq 3\|T_g f - \widetilde{f}\|_{2,\mu}^2 + \frac{2C_1 \log(4|\mathcal{F}|/\delta)}{N}$$

$$= 3\min_{h \in \mathcal{F}} \|h - T_g f\|_{2,\mu}^2 + \frac{2C_1 \log(4|\mathcal{F}|/\delta)}{N} \leq 3\varepsilon_c + \frac{2C_1 \log(4|\mathcal{F}|/\delta)}{N}, \quad (20)$$

where the last inequality follows from the approximate robust Bellman completion assumption (Assumption 2).

We note that since $\widehat{f}_g$ is the least-squares regression solution, we know that $(1/N) \sum_{i=1}^N z_i^{\widehat{f}_g} \leq (1/N) \sum_{i=1}^N z_i^{\widetilde{f}}$. With this note in (20), from (16), with probability at least $1 - \delta$, we have

$$\|T_g f - \widehat{f}_g\|_{2,\mu}^2 \leq 3\varepsilon_c + \frac{2C_1 \log(4|\mathcal{F}|/\delta)}{N}$$

$$+ \sqrt{\frac{2C_1 \|T_g f - \widehat{f}_g\|_{2,\mu}^2 \log(4|\mathcal{F}|/\delta)}{N}} + \frac{2C_1 \log(4|\mathcal{F}|/\delta)}{3N}$$

$$\leq 3\varepsilon_c + \frac{3C_1 \log(4|\mathcal{F}|/\delta)}{N} + \sqrt{\frac{3C_1 \|T_g f - \widehat{f}_g\|_{2,\mu}^2 \log(4|\mathcal{F}|/\delta)}{N}}.$$

From the earlier algebra fact, taking $x = \|T_g f - \widehat{f}_g\|_{2,\mu}^2$, with probability at least $1 - \delta$, we have

$$\|T_g f - \widehat{f}_g\|_{2,\mu}^2 \leq 6\varepsilon_c + \frac{9C_1 \log(4|\mathcal{F}|/\delta)}{N}.$$

From the fact $\sqrt{x+y} \leq \sqrt{x} + \sqrt{y}$, with probability at least $1 - \delta$, we get

$$\|T_g f - \widehat{f}_g\|_{2,\mu} \leq \sqrt{6\varepsilon_c} + \sqrt{\frac{9C_1 \log(4|\mathcal{F}|/\delta)}{N}}.$$

Using union bound for $f \in \mathcal{F}$ and $g \in \mathcal{G}$, with probability at least $1 - \delta$, we finally obtain

$$\sup_{f \in \mathcal{F}} \sup_{g \in \mathcal{G}} \|T_g f - \widehat{f}_g\|_{2,\mu} \leq \sqrt{6\varepsilon_c} + \sqrt{\frac{18 C_1 \log(2|\mathcal{F}||\mathcal{G}|/\delta)}{N}},$$

which completes the least-squares generalization bound analysis. $\qquad\square$

We are now ready to prove the main theorem.

***Proof of Theorem 1.*** We let $V_k(s) = Q_k(s, \pi_k(s))$ for every $s \in \mathcal{S}$. Since $\pi_k$ is the greedy policy w.r.t $Q_k$, we also have $V_k(s) = Q_k(s, \pi_k(s)) = \max_a Q_k(s, a)$. We recall that $V^* = V^{\pi^*}$ and $Q^* = Q^{\pi^*}$. We also recall from Section 2 that $Q^{\pi^*}$ is a fixed-point of the robust Bellman operator $T$ defined in (3). We also note that the same holds true for any stationary deterministic policy $\pi$ from Iyengar (2005) that $Q^\pi$ satisfies $Q^\pi(s, a) = r(s, a) + \gamma \min_{P_{s,a} \in \mathcal{P}_{s,a}} \mathbb{E}_{s' \sim P_{s,a}}[V^\pi(s')]$. We can now further use the dual form (5) under Assumption 3. We first characterize the performance decomposition between $V^{\pi^*}$ and $V^{\pi_K}$. For a given $s_0 \in \mathcal{S}$, we observe that

$$V^{\pi^*}(s_0) - V^{\pi_K}(s_0) = (V^{\pi^*}(s_0) - V_K(s_0)) - (V^{\pi_K}(s_0) - V_K(s_0))$$

$$= (Q^{\pi^*}(s_0, \pi^*(s_0)) - Q_K(s_0, \pi_K(s_0))) - (Q^{\pi_K}(s_0, \pi_K(s_0)) - Q_K(s_0, \pi_K(s_0)))$$

$$\overset{(a)}{\leq} Q^{\pi^*}(s_0, \pi^*(s_0)) - Q_K(s_0, \pi^*(s_0)) + Q_K(s_0, \pi_K(s_0)) - Q^{\pi_K}(s_0, \pi_K(s_0))$$

$$= Q^{\pi^*}(s_0, \pi^*(s_0)) - Q_K(s_0, \pi^*(s_0)) + Q_K(s_0, \pi_K(s_0)) - Q^{\pi^*}(s_0, \pi_K(s_0))$$

$$\qquad\qquad + Q^{\pi^*}(s_0, \pi_K(s_0)) - Q^{\pi_K}(s_0, \pi_K(s_0))$$

$$\overset{(b)}{\leq} Q^{\pi^*}(s_0, \pi^*(s_0)) - Q_K(s_0, \pi^*(s_0)) + Q_K(s_0, \pi_K(s_0)) - Q^{\pi^*}(s_0, \pi_K(s_0))$$

$$\qquad\qquad + \gamma \sup_\eta (\mathbb{E}_{s_1 \sim P^o_{s_0, \pi_K(s_0)}}((\eta - V^{\pi_K}(s_1))_+ - (\eta - V^{\pi^*}(s_1))_+))$$

$$\overset{(c)}{\leq} |Q^{\pi^*}(s_0, \pi^*(s_0)) - Q_K(s_0, \pi^*(s_0))| + |Q^{\pi^*}(s_0, \pi_K(s_0)) - Q_K(s_0, \pi_K(s_0))|$$

$$\qquad\qquad + \gamma \mathbb{E}_{s_1 \sim P^o_{s_0, \pi_K(s_0)}}(|V^{\pi^*}(s_1) - V^{\pi_K}(s_1)|).$$

$(a)$ follows from the fact that $\pi_K$ is the greedy policy with respect to $Q_K$. $(b)$ follows from the Bellman optimality equations and the fact $|\sup_x f(x) - \sup_x g(x)| \leq \sup_x |f(x) - g(x)|$. Finally, $(c)$ follows from the facts $(x)_+ - (y)_+ \leq (x - y)_+$ and $(x)_+ \leq |x|$ for any $x, y \in \mathbb{R}$.

We now recall the initial state distribution $d_0$. Thus, we have

$$\mathbb{E}_{s_0 \sim d_0}[V^{\pi^*}] - \mathbb{E}_{s_0 \sim d_0}[V^{\pi_K}] \leq$$

$$\mathbb{E}_{s_0 \sim d_0}\Bigg[|Q^{\pi^*}(s_0, \pi^*(s_0)) - Q_K(s_0, \pi^*(s_0))| + |Q^{\pi^*}(s_0, \pi_K(s_0)) - Q_K(s_0, \pi_K(s_0))|$$

$$\qquad\qquad + \gamma \mathbb{E}_{s_1 \sim P^o_{s_0, \pi_K(s_0)}}(|V^{\pi^*}(s_1) - V^{\pi_K}(s_1)|)\Bigg].$$

Since $V^{\pi^*}(s) \geq V^{\pi_K}(s)$ for any $s \in \mathcal{S}$, by telescoping we get

$$\mathbb{E}_{s_0 \sim d_0}[V^{\pi^*}] - \mathbb{E}_{s_0 \sim d_0}[V^{\pi_K}] \leq \sum_{h=0}^\infty \gamma^h \times$$

$$\left(\mathbb{E}_{s \sim d_{h, \pi_K}}[|Q^{\pi^*}(s, \pi^*(s)) - Q_K(s, \pi^*(s))| + |Q^{\pi^*}(s, \pi_K(s)) - Q_K(s, \pi_K(s))|]\right), \quad (21)$$

where $d_{h, \pi_K} \in \Delta(\mathcal{S})$ for all natural numbers $h \geq 0$ is defined as

$$d_{h, \pi_K} = \begin{cases} d_0 & \text{if } h = 0, \\ P^o_{s', \pi_K(s')} & \text{otherwise, with } s' \sim d_{h-1, \pi_K}. \end{cases}$$

We emphasize that the state distribution $d_{h, \pi_K}$'s are different from the discounted state-action occupancy distributions. We note that a similar state distribution proof idea is used in Agarwal et al. (2019).

Recall $\|f\|_{p,\nu}^2 = (\mathbb{E}_{s,a\sim\nu}|f(s,a)|^p)^{1/p}$, where $\nu \in \Delta(\mathcal{S}\times\mathcal{A})$. With this we have

$$\mathbb{E}_{s_0\sim d_0}[V^{\pi^*}] - \mathbb{E}_{s_0\sim d_0}[V^{\pi_K}] \leq \sum_{h=0}^{\infty}\gamma^h\bigg(\|Q^{\pi^*} - Q_K\|_{1,d_{h,\pi_K}\circ\pi^*} + \|Q^{\pi^*} - Q_K\|_{1,d_{h,\pi_K}\circ\pi_K}\bigg),$$
(22)

where the state-action distributions $d_{h,\pi_K}\circ\pi^*(s,a) \propto d_{h,\pi_K}(s)\mathbb{1}\{a = \pi^*(s)\}$ and $d_{h,\pi_K}\circ\pi_K(s,a) \propto d_{h,\pi_K}(s)\mathbb{1}\{a = \pi_K(s)\}$ directly follows by comparing with (21).

We now bound one of the RHS terms above by bounding for any state-action distribution $\nu$ satisfying Assumption 1 (in particular the following bound is true for $d_{h,\pi_K}\circ\pi^*$ or $d_{h,\pi_K}\circ\pi_K$ in (21)):

$$\|Q^{\pi^*} - Q_K\|_{1,\nu} \leq \|Q^{\pi^*} - TQ_{K-1}\|_{1,\nu} + \|TQ_{K-1} - Q_K\|_{1,\nu}$$

$$\overset{(a)}{\leq} \|Q^{\pi^*} - TQ_{K-1}\|_{1,\nu} + \sqrt{C}\|TQ_{K-1} - Q_K\|_{1,\mu}$$

$$= (\mathbb{E}_{s,a\sim\nu}|Q^{\pi^*}(s,a) - TQ_{K-1}(s,a)|) + \sqrt{C}\|TQ_{K-1} - Q_K\|_{1,\mu}$$

$$\overset{(b)}{\leq} (\mathbb{E}_{s,a\sim\nu}\gamma\sup_{\eta}|\mathbb{E}_{s'\sim P^o_{s,a}}((\eta - \max_{a'}Q_{K-1}(s',a'))_+ - (\eta - \max_{a'}Q^{\pi^*}(s',a'))_+)|)$$

$$\qquad\qquad\qquad\qquad + \sqrt{C}\|TQ_{K-1} - Q_K\|_{1,\mu}$$

$$\overset{(c)}{\leq} (\mathbb{E}_{s,a\sim\nu}|\mathbb{E}_{s'\sim P^o_{s,a}}(\max_{a'}Q^{\pi^*}(s',a') - \max_{a'}Q_{K-1}(s',a'))_+|) + \sqrt{C}\|TQ_{K-1} - Q_K\|_{1,\mu}$$

$$\overset{(d)}{\leq} \gamma(\mathbb{E}_{s,a\sim\nu}\mathbb{E}_{s'\sim P^o_{s,a}}\max_{a'}|Q^{\pi^*}(s',a') - Q_{K-1}(s',a')|) + \sqrt{C}\|TQ_{K-1} - Q_K\|_{1,\mu}$$

$$\overset{(e)}{\leq} \gamma\|Q^{\pi^*} - Q_{K-1}\|_{1,\nu'} + \sqrt{C}\|TQ_{K-1} - Q_K\|_{1,\mu}$$

$$\overset{(f)}{\leq} \gamma\|Q^{\pi^*} - Q_{K-1}\|_{1,\nu'} + \sqrt{C}\|T_{g_{K-1}}Q_{K-1} - Q_K\|_{2,\mu} + \sqrt{C}\|TQ_{K-1} - T_{g_{K-1}}Q_{K-1}\|_{1,\mu},$$
(23)

where $(a)$ follows by the concentratability assumption (Assumption 1), $(b)$ from Bellman equation, operator $T$, and the fact $|\sup_x p(x) - \sup_x q(x)| \leq \sup_x |p(x) - q(x)|$, $(c)$ from the fact $|(x)_+ - (y)_+| \leq |(x-y)_+|$ for any $x, y \in \mathbb{R}$, $(d)$ follows by Jensen's inequality and by the facts $|\sup_x p(x) - \sup_x q(x)| \leq \sup_x |p(x) - q(x)|$ and $(x)_+ \leq |x|$ for any $x, y \in \mathbb{R}$, and $(e)$ by defining the distribution $\nu'$ as $\nu'(s',a') = \sum_{s,a}\nu(s,a)P^o_{s,a}(s')\mathbb{1}\{a' = \arg\max_b |Q^{\pi^*}(s',b) - Q_{K-1}(s',b)|\}$, and $(f)$ using the fact that $\|\cdot\|_{1,\mu} \leq \|\cdot\|_{2,\mu}$.

Now, by recursion until iteration 0, we get

$$\|Q^{\pi^*} - Q_K\|_{1,\nu} \leq \gamma^K\sup_{\bar{\nu}}\|Q^{\pi^*} - Q_0\|_{1,\bar{\nu}} + \sqrt{C}\sum_{t=0}^{K-1}\gamma^t\|TQ_{K-1-t} - T_{g_{K-1-t}}Q_{K-1-t}\|_{1,\mu}$$

$$+ \sqrt{C}\sum_{t=0}^{K-1}\gamma^t\|T_{g_{K-1-t}}Q_{K-1-t} - Q_{K-t}\|_{2,\mu}$$

$$\overset{(a)}{\leq} \frac{\gamma^K}{1-\gamma} + \sqrt{C}\sum_{t=0}^{K-1}\gamma^t\|TQ_{K-1-t} - T_{g_{K-1-t}}Q_{K-1-t}\|_{1,\mu}$$

$$+ \sqrt{C}\sum_{t=0}^{K-1}\gamma^t\|T_{g_{K-1-t}}Q_{K-1-t} - Q_{K-t}\|_{2,\mu}$$

$$\overset{(b)}{\leq} \frac{\gamma^K}{1-\gamma} + \frac{\sqrt{C}}{1-\gamma}\sup_{f\in\mathcal{F}}\|Tf - T_{\widehat{g}_f}f\|_{1,\mu} + \frac{\sqrt{C}}{1-\gamma}\sup_{f\in\mathcal{F}}\|T_{\widehat{g}_f}f - \widehat{f}_{\widehat{g}_f}\|_{2,\mu}$$

$$\leq \frac{\gamma^K}{1-\gamma} + \frac{\sqrt{C}}{1-\gamma}\sup_{f\in\mathcal{F}}\|Tf - T_{\widehat{g}_f}f\|_{1,\mu} + \frac{\sqrt{C}}{1-\gamma}\sup_{f\in\mathcal{F}}\sup_{g\in\mathcal{G}}\|T_g f - \widehat{f}_g\|_{2,\mu}.$$
(24)

where $(a)$ follows since $|Q^{\pi^*}(s,a)| \leq 1/(1-\gamma)$, $Q_0(s,a) = 0$, and $(b)$ follows since $\widehat{g}_f$ is the dual variable function from the algorithm for the state-action value function $f$ and $\widehat{f}_g$ as the least squares solution from the algorithm for the state-action value function $f$ and dual variable function $g$ pair.

The proof is now complete combining (22) and (24) with Lemma 6 and Lemma 7. □

# D  Related Works

Here we provide a more detailed description of the related work to complement what we listed in the introduction (Section 1).

**Offline RL:** The problem of learning the optimal policy only using an offline dataset is first addressed under the generative model assumption (Singh and Yee, 1994; Azar et al., 2013; Haskell et al., 2016; Sidford et al., 2018; Agarwal et al., 2020; Li et al., 2020; Kalathil et al., 2021). This assumption requires generating the same uniform number of next-state samples for each and every state-action pairs. To account for large state spaces, there are number of works (Antos et al., 2008; Bertsekas, 2011; Lange et al., 2012; Chen and Jiang, 2019; Xie and Jiang, 2020; Levine et al., 2020; Xie et al., 2021) that utilize function approximation under similar assumption, concentratability assumption (Chen and Jiang, 2019) in which the data distribution $\mu$ sufficiently covers the discounted state-action occupancy. There is rich literature (Munos and Szepesvári, 2008; Farahmand et al., 2010; Lazaric et al., 2012; Chen and Jiang, 2019; Liu et al., 2020; Xie et al., 2021) in the conquest of identifying and improving these necessary and sufficient assumptions for offline RL that use variations of Fitted Q-Iteration (FQI) algorithm (Gordon, 1995; Ernst et al., 2005). There is also rich literature (Fujimoto et al., 2019; Kumar et al., 2019, 2020; Yu et al., 2020; Zhang and Jiang, 2021) that develop offline deep RL algorithms focusing on the algorithmic and empirical aspects and propose multitude heuristic approaches to advance the field. All these results assume that the offline data is generated according to a single model and the goal is to find the optimal policy for the MDP with the same model. In particular, none of these works consider the *offline robust RL problem* where the offline data is generated according to a (training) model which can be different from the one in testing, and the goal is to learn a policy that is robust w.r.t. an uncertainty set.

**Robust RL:** To address the parameter uncertainty problem, Iyengar (2005) and Nilim and El Ghaoui (2005) introduced the RMDP framework. Iyengar (2005) showed that the optimal robust value function and policy can be computed using the robust counterparts of the standard value iteration and policy iteration algorithms. To tackle the parameter uncertainty problem, other works considered distributionally robust setting (Xu and Mannor, 2010), modified policy iteration (Kaufman and Schaefer, 2013), and more general uncertainty set (Wiesemann et al., 2013). These initial works mainly focused on the planning problem (known transition probability dynamics) in the tabular setting. Tamar et al. (2014) proposed linear function approximation method to solve large RMDPs. Though this work suggests a sampling based approach, a general model-free learning algorithm and analysis was not included. Roy et al. (2017) proposed the robust versions of the classical model-free reinforcement learning algorithms, such as Q-learning, SARSA, and TD-learning in the tabular setting. They also proposed function approximation based algorithms for the policy evaluation. However, this work does not have a policy iteration algorithm with provable guarantees for learning the optimal robust policy. Derman et al. (2018) introduced soft-robust actor-critic algorithms using neural networks, but does not provide any global convergence guarantees for the learned policy. Tessler et al. (2019) proposed a min-max game framework to address the robust learning problem focusing on the tabular setting. Lim and Autef (2019) proposed a kernel-based RL algorithm for finding the robust value function in a batch learning setting. Mankowitz et al. (2020) employed an entropy-regularized policy optimization algorithm for continuous control using neural network, but does not provide any provable guarantees for the learned policy. Panaganti and Kalathil (2021) proposed least-squares policy iteration method to handle large state-action space in robust RL, but only provide asymptotic policy evaluation convergence guarantees whereas Panaganti and Kalathil (2021) provide finite time convergence for the policy iteration to optimal robust value.

**Other robust RL related works:** Robust control is a well-studied area in the classical control theory (Zhou et al., 1996; Dullerud and Paganini, 2013). Recently, there are some interesting works that address the robust RL problem using this framework, especially focusing on the linear quadratic regulator setting (Zhang et al., 2020b). Risk sensitive RL algorithms (Borkar, 2002; Prashanth and Ghavamzadeh, 2016; Fei et al., 2021) and adversarial RL algorithms (Pinto et al., 2017; Zhang et al., 2021; Huang et al., 2022) also address the robustness problem implicitly under different frameworks which are independent from RMDPs. (Zhang et al., 2022) addresses the problem of *corruption-robust* offline RL problem, where an adversary is allowed to change a fraction of the samples of an offline RL dataset and the goal is to find the optimal policy for the nominal linear MDP model (according to

which the offline data is generated). Our framework and approach of robust MDP is significantly different from these line of works.

**This work:** The works that are closest to ours are by Zhou et al. (2021); Yang et al. (2021); Panaganti and Kalathil (2022) that address the robust RL problem in a tabular setting under the generative model assumption. Due to the generative model assumption, the offline data has the same uniform number of samples corresponding to each and every state-action pair, and tabular setting allows the estimation of the uncertainty set followed by solving the planning problem. Our work is significantly different from these in the following way: $(i)$ we consider a robust RL problem with arbitrary large state space, instead of the small tabular setting, $(ii)$ we consider a true offline RL setting where the state-action pairs are sampled according to an arbitrary distribution, instead of using the generative model assumption, $(iii)$ we focus on a function approximation approach where the goal is to directly learn optimal robust value/policy using function approximation techniques, instead of solving the tabular planning problem with the estimated model. *To the best of our knowledge, this is the first work that addresses the offline robust RL problem with arbitrary large state space using function approximation, with provable guarantees on the performance of the learned policy.*

# E    Experiment Details

We provide more detailed and practical version of our RFQI algorithm (Algorithm 1) in this section. We also provide more experimental results evaluated on *Cartpole*, *Hopper*, and *Half-Cheetah* OpenAI Gym Mujoco (Brockman et al., 2016) environments.

We provide our code in **github webpage** `https://github.com/zaiyan-x/RFQI` containing instructions to reproduce all results in this paper. We implemented our RFQI algorithm based on the architecture of Batch Constrained deep Q-learning (BCQ) algorithm (Fujimoto et al., 2019) [2] and Pessimistic Q-learning (PQL) algorithm (Liu et al., 2020) [3]. We note that PQL algorithm (with $b = 0$ filtration thresholding (Liu et al., 2020)) and BCQ algorithm are the practical versions of FQI algorithm with neural network architecture.

## E.1    RFQI Practical Algorithm

We provide the practical version of our RFQI algorithm in Algorithm 2 and highlight the difference with BCQ and PQL algorithms in blue (steps 8 and 9).

**RFQI algorithm implementation details**: The Variational Auto-Encoder (VAE) $G_\omega^a$ (Kingma and Welling, 2013) is defined by two networks, an encoder $E_{\omega_1}(s, a)$ and decoder $D_{\omega_2}(s, z)$, where $\omega = \{\omega_1, \omega_2\}$. The encoder outputs mean and standard deviation, $(\mu, \sigma) = E_{\omega_1}(s, a)$, of a normal distribution. A latent vector $z$ is sampled from the standard normal distribution and for a state $s$, the decoder maps them to an action $D_{\omega_2} : (s, z) \mapsto \tilde{a}$. Then the evidence lower bound ($ELBO$) of VAE is given by $ELBO(B; G_\omega^a) = \sum_B (a - \tilde{a})^2 + D_{\text{KL}}(\mathcal{N}(\mu, \sigma), \mathcal{N}(0, 1))$, where $\mathcal{N}$ is the normal distribution with mean and standard deviation parameters. We refer to (Fujimoto et al., 2019) for more details on VAE. We also use the default VAE architecture from BCQ algorithm (Fujimoto et al., 2019) and PQL algorithm (Liu et al., 2020) in our RFQI algorithm.

We now focus on the additions described in blue (steps 8 and 9) in Algorithm 2. For all the other networks we use default architecture from BCQ algorithm (Fujimoto et al., 2019) and PQL algorithm (Liu et al., 2020) in our RFQI algorithm.
(1) In each iteration $k$, we solve the dual variable function $g_\theta$ optimization problem (step 4 in Algorithm 1, step 8 in Algorithm 2) implemented by ADAM (Kingma and Ba, 2014) on the minibatch $B$ with the learning rate $l_1$ mentioned in Table 1.
(2) Our state-action value target function corresponds to the robust state-action value target function described in (10). This is reflected in step 9 of Algorithm 2. The state-action value function $Q_\theta$ optimization problem (step 5 in Algorithm 1, step 9 in Algorithm 2) is implemented by ADAM (Kingma and Ba, 2014) on the minibatch $B$ with the learning rate $l_2$ mentioned in Table 1.

---

[2] Available at `https://github.com/sfujim/BCQ`
[3] Available at `https://github.com/yaoliucs/PQL`

---

**Algorithm 2** RFQI Practical Algorithm

---

1: **Input:** Offline dataset $\mathcal{D}$, radius of robustness $\rho$, maximum perturbation $\Phi$, target update rate $\tau$, mini-batch size $N$, maximum number of iterations $K$, number of actions $u$.
2: **Initialize:** Two state-action neural networks $Q_{\theta_1}$ and $Q_{\theta_2}$, one dual neural network $g_{\theta_3}$ policy (perturbation) model: $\xi_\varphi \in [-\Phi, \Phi])$, and action VAE $G_\omega^a$, with random parameters $\theta_1$, $\theta_2$, $\varphi$, $\omega$, and target networks $Q_{\theta_1'}, Q_{\theta_2'}, \xi_{\varphi'}$ with $\theta_1' \leftarrow \theta_1, \theta_2' \leftarrow \theta_2, \varphi' \leftarrow \varphi$.
3: **for** $k = 1, \cdots, K$ **do**
4:     Sample a minibatch $B$ with $N$ samples from $\mathcal{D}$.
5:     Train $\omega \leftarrow \arg\min_\omega ELBO(B; G_\omega^a)$. Sample $u$ actions $a_i'$ from $G_\omega^a(s')$ for each $s'$.
6:     Perturb $u$ actions $a_i' = a_i' + \xi_\varphi(s', a_i')$.
7:     Compute next-state value target for each $s'$ in $B$:

$$V_t = \max_{a_i'}(0.75 \cdot \min\{Q_{\theta_1'}, Q_{\theta_2'}\} + 0.25 \cdot \max\{Q_{\theta_1'}, Q_{\theta_2'}\}).$$

8:     $\theta_3 \leftarrow \arg\min_\theta \sum[\max\{g_\theta(s, a) - V_t(s'), 0\} - (1 - \rho)g_\theta(s, a)].$
9:     Compute next-state Q target for each $(s, a, r, s')$ pair in $B$:

$$Q_t(s, a) = r - \gamma \cdot \max\{g_{\theta_3}(s, a) - V_t(s'), 0\} + \gamma(1 - \rho)g_{\theta_3}(s, a).$$

10:     $\theta \leftarrow \arg\min_\theta \sum(Q_t(s, a) - Q_\theta(s, a))^2.$
11:     Sample $u$ actions $a_i$ from $G_\omega^a(s)$ for each $s$.
12:     $\varphi \leftarrow \arg\max_\varphi \sum \max_{a_i} Q_{\theta_1}(s, a_i + \xi_\varphi(s, a_i)).$
13:     Update target network: $\theta' = (1 - \tau)\theta' + \tau\theta, \varphi' = (1 - \tau)\varphi' + \tau\varphi.$
14: **end for**
15: **Output policy:** Given $s$, sample $u$ actions $a_i$ from $G_\omega^a(s)$. Select action $a = \arg\max_{a_i} Q_{\theta_1}(s, a_i + \xi_\varphi(s, a_i)).$

---

| Environment | Discount $\gamma$ | Learning rates $[l_1, l_2]$ | Q Neural nets $\theta_1 = \theta_2 = [h_1, h_2]$ | Dual Neural nets $\theta_3 = [h_1, h_2]$ |
|---|---|---|---|---|
| CartPole | 0.99 | $[10^{-3}, 10^{-3}]$ | $[400, 300]$ | $[64, 64]$ |
| Hopper | 0.99 | $[10^{-3}, 8 \times 10^{-4}]$ $[3 \times 10^{-4}, 6 \times 10^{-4}]$ | $[400, 300]$ | $[64, 64]$ |
| Half-Cheetah | 0.99 | $[10^{-3}, 8 \times 10^{-4}]$ $[3 \times 10^{-4}, 6 \times 10^{-4}]$ | $[400, 300]$ | $[64, 64]$ |

Table 1: Details of hyper-parameters in FQI and RFQI algorithms experiments.

**Hyper-parameters details**: We now give the description of hyper-parameters used in our codebase in Table 1. We use same hyper-parameters across different algorithms. Across all learning algorithms we use $\tau = 0.005$ for the target network update, $K = 5 \times 10^5$ for the maximum iterations, $|\mathcal{D}| = 10^6$ for the offline dataset, $|B| = 1000$ for the minibatch size. We used grid-search for $\rho$ in $\{0.2, 0.3, \cdots, 0.6\}$. We also picked best of the two sets of learning rates mentioned in Table 1. For all the other hyper-parameters we use default values from BCQ algorithm (Fujimoto et al., 2019) and PQL algorithm (Liu et al., 2020) in our RFQI algorithm that can be found in our code.

**Offline datasets**: Now we discuss the offline dataset used in the our training of FQI and RFQI algorithms. For the fair comparison in every plot, we train both FQI and RFQI algorithms on same offline datasets.

*Cartpole dataset $\mathcal{D}_c$*: We first train proximal policy optimization (PPO) (Schulman et al., 2017) algorithm, under default RL baseline zoo (Raffin, 2020) parameters. We then generate the Cartpole dataset $\mathcal{D}_c$ with $10^5$ samples using an $\varepsilon$-greedy ($\varepsilon = 0.3$) version of this PPO trained policy. We note that this offline dataset contains non-expert behavior meeting the richness of the data-generating distribution assumption in practice.

*Mixed dataset $\mathcal{D}_m$*: For the MuJoCo environments, *Hopper* and *Half-Cheetah*, we increase the richness of the dataset since these are high dimensional problems. We first train soft actor-critic (SAC) (Haarnoja et al., 2018) algorithm, under default RL baseline zoo (Raffin, 2020) parameters, with replay

buffer updated by a fixed $\varepsilon$-greedy ($\varepsilon = 0.1$) policy with the model parameter *actuator_ctrlrange* set to $[-0.85, 0.85]$. We then generate the mixed dataset $\mathcal{D}_m$ with $10^6$ samples from this $\varepsilon$-greedy ($\varepsilon = 0.3$) SAC trained policy. We note that such a dataset generation gives more diverse set of observations than the process of $\mathcal{D}_c$ generation for fair comparison between FQI and RFQI algorithms.

*D4RL dataset* $\mathcal{D}_d$: We consider the *hopper-medium* and *halfcheetah-medium* offline datasets in (Fu et al., 2020) which are benchmark datasets in offline RL literature (Fu et al., 2020; Levine et al., 2020; Liu et al., 2020). These 'medium' datasets are generated by first training a policy online using Soft Actor-Critic (Haarnoja et al., 2018), early-stopping the training, and collecting $10^6$ samples from this partially-trained policy. We refer to (Fu et al., 2020) for more details.

We end this section by mentioning the software and hardware configurations used. The training and evaluation is done using three computers with the following configuration. Operating system is Ubuntu 18.04 and Lambda Stack; main softwares are PyTorch, Caffe, CUDA, cuDNN, Numpy, Matplotlib; processor is AMD Threadripper 3960X (24 Cores, 3.80 GHz); GPUs are 2x RTX 2080 Ti; memory is 128GB RAM; Operating System Drive is 1 TB SSD (NVMe); and Data Drive is 4TB HDD.

## E.2 More Experimental Results

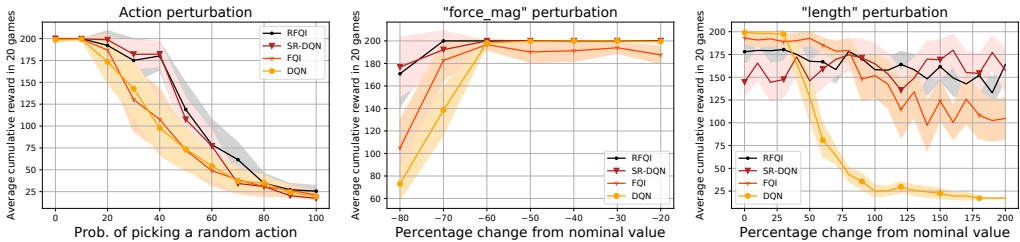

Figure 4: *Cartpole simulation results on offline dataset $\mathcal{D}_c$.* Average cumulative reward in 20 episodes versus different model parameter perturbations mentioned in the respective titles.

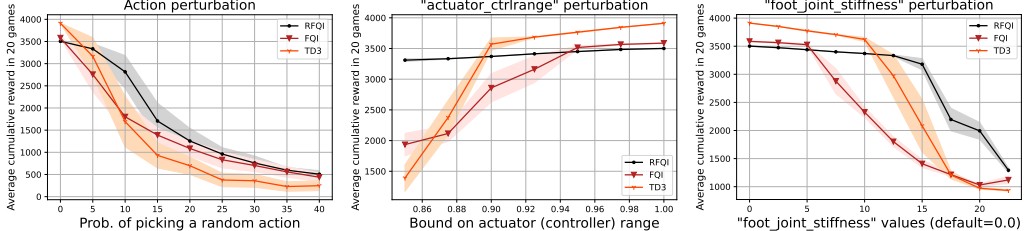

Figure 5: *Hopper simulation results on offline dataset $\mathcal{D}_m$.* Average cumulative reward in 20 episodes versus different model parameter perturbations mentioned in the respective titles.

Here we provide more experimental results and details in addition to Fig. 1-3 in Section 5.

For the *Cartpole*, we compare RFQI algorithm against the non-robust RL algorithms FQI and DQN, and the soft-robust RL algorithm proposed in Derman et al. (2018). We trained FQI and RFQI algorithms on the dataset $\mathcal{D}_c$ (a detailed description of data set is provided in Appendix E.1). We test the robustness of the algorithms by changing the parameters *force_mag* (to model external force disturbance), *length* (to model change in pole length), and also by introducing action perturbations (to model actuator noise). The nominal value of *force_mag* and *length* parameters are 10 and 0.5 respectively. Fig. 4 shows superior robust performance of RFQI compared to the non-robust FQI and DQN. For example, consider the action perturbation performance plot in Fig. 4 where RFQI algorithm improves by $75\%$ compared to FQI algorithm in average cumulative reward for a $40\%$ chance of action perturbation. We note that we found $\rho = 0.5$ is the best from grid-search for RFQI algorithm. The RFQI performance is similar to that of soft-robust DQN. We note that soft-robust DQN algorithm is an online deep RL algorithm (and not an offline RL algorithm) and has no provable performance guarantee. Moreover, soft-robust DQN algorithm requires generating online data according a number of models in the uncertainty set, whereas RFQI only requires offline data according to a single nominal training model.

Before we proceed to describe our results on the OpenAI Gym MuJoCo (Brockman et al., 2016) environments *Hopper* and *Half-Cheetah*, we first mention their model parameters and its corresponding nominal values in Table 2. The model parameter names are self-explanatory, for example, stiffness control on the leg joint is the *leg_joint_stiffness*, range of actuator values is the *actuator_ctrlrange*. The front and back parameters in Half-Cheetah are for the front and back legs. We refer to the perturbed environments provided in our code and the *hopper.xml, halfcheetah.xml* files in the environment assets of OpenAI Gym MuJoCo (Brockman et al., 2016) for more information regarding these model parameters.

| Environment | Model parameter | Nominal range/value |
|---|---|---|
| Hopper | *actuator_ctrlrange* | $[-1, 1]$ |
| | *foot_joint_stiffness* | 0 |
| | *leg_joint_stiffness* | 0 |
| | *thigh_joint_stiffness* | 0 |
| | *joint_damping* | 1 |
| | *joint_frictionloss* | 0 |
| Half-Cheetah | *joint_frictionloss* | 0 |
| | front *actuator_ctrlrange* | $[-1, 1]$ |
| | back *actuator_ctrlrange* | $[-1, 1]$ |
| | front *joint_stiffness* = (*thigh_joint_stiffness*, *shin_joint_stiffness*, *foot_joint_stiffness*) | $(180, 120, 60)$ |
| | back *joint_stiffness* = (*thigh_joint_stiffness*, *shin_joint_stiffness*, *foot_joint_stiffness*) | $(240, 180, 120)$ |
| | front *joint_damping* = (*thigh_joint_damping*, *shin_joint_damping*, *foot_joint_damping*) | $(4.5, 3.0, 1.5)$ |
| | back *joint_damping* = (*thigh_joint_damping*, *shin_joint_damping*, *foot_joint_damping*) | $(6.0, 4.5, 3.0)$ |

Table 2: Details of model parameters for *Hopper* and *Half-Cheetah* environments.

For the *Hopper*, we compare RFQI algorithm against the non-robust RL algorithms FQI and TD3 (Fujimoto et al., 2018). We trained FQI and RFQI algorithms on the mixed dataset $\mathcal{D}_m$ (a detailed description of dataset provided in Appendix E.1). We note that we do not compare with soft robust RL algorithms because of its poor performance on MuJoCo environments in the rest of our figures. We test the robustness of the algorithm by introducing action perturbations, and by changing the model parameters *actuator_ctrlrange*, *foot_joint_stiffness*, and *leg_joint_stiffness*. Fig. 3 and Fig. 5 shows RFQI algorithm is consistently robust compared to the non-robust algorithms. We note that we found $\rho = 0.5$ is the best from grid-search for RFQI algorithm. The average episodic reward of RFQI remains almost the same initially, and later decays much less and gracefully when compared to FQI and TD3 algorithms. For example, in plot 3 in Fig. 5, at the *foot_joint_stiffness* parameter value 15, the episodic reward of FQI is only around 1400 whereas RFQI achieves an episodic reward of 3200. Similar robust performance of RFQI can be seen in other plots as well. We also note that TD3 (Fujimoto et al., 2019) is a powerful off-policy policy gradient algorithm that relies on large $10^6$ replay buffer of online data collection, unsurprisingly performs well initially with less perturbation near the nominal models.

In order to verify the effectiveness and consistency of our algorithm across different offline dataset, we repeat the above experiments, on additional OpenAI Gym MuJoCo (Brockman et al., 2016) environment *Half-Cheetah*, using D4RL dataset $\mathcal{D}_d$ (a detailed description of dataset provided in Appendix E.1) which are benchmark in offline RL literature (Fu et al., 2020; Levine et al., 2020; Liu et al., 2020) than our mixed dataset $\mathcal{D}_m$. Since D4RL dataset is a benchmark dataset for offline RL algorithms, here we focus only on the comparison between the two offline RL algorithms we consider, our RFQI algorithm and its non-robust counterpart FQI algorithm. We now showcase the results on *Hopper* and *Half-Cheetah* for this setting.

For the *Hopper*, we test the robustness by changing the model parameters *gravity*, *joint_damping*, and *joint_frictionloss*. Fig. 6 shows RFQI algorithm is consistently robust compared to the non-robust FQI algorithm. We note that we found $\rho = 0.5$ is the best from grid-search for RFQI algorithm. The average episodic reward of RFQI remains almost the same initially, and later decays much less and gracefully when compared to FQI algorithm. For example, in plot 2 in Fig. 6, for the 30%

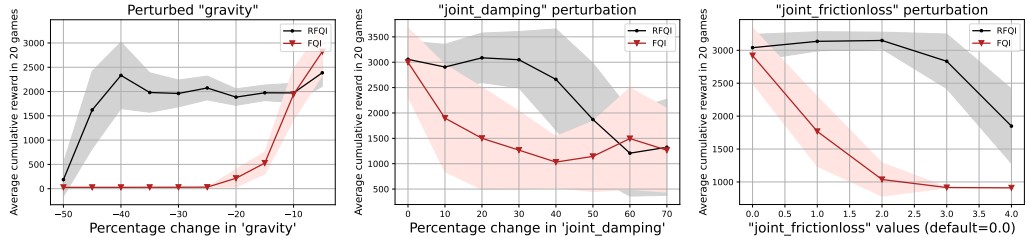

Figure 6: *Hopper evaluation simulation results on offline dataset $\mathcal{D}_d$.* Average cumulative reward in 20 episodes versus different model parameter perturbations mentioned in the respective titles.

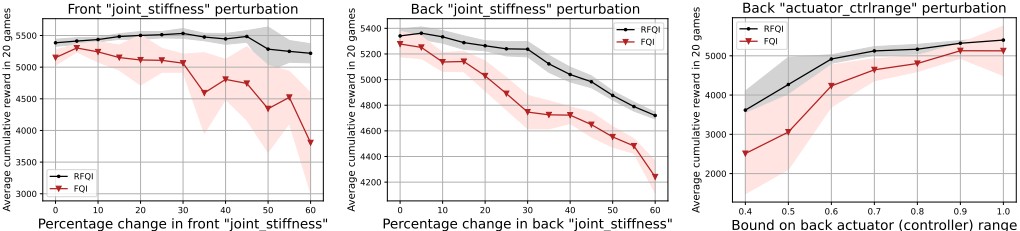

Figure 7: *Half-Cheetah evaluation simulation results on offline dataset $\mathcal{D}_d$.* Average cumulative reward in 20 episodes versus different model parameter perturbations mentioned in the respective titles.

change in *joint_damping* parameter, the episodic reward of FQI is only around 1400 whereas RFQI achieves an episodic reward of 3000 which is almost the same as for unperturbed model. Similar robust performance of RFQI can be seen in other plots as well.

For the *Half-Cheetah*, we test the robustness by changing the model parameters *joint_stiffness* of front and back joints, and *actuator_ctrlrange* of back joint. Fig. 7 shows RFQI algorithm is consistently robust compared to the non-robust FQI algorithm. We note that we found $\rho = 0.3$ is the best from grid-search for RFQI algorithm. For example, in plot 1 in Fig. 7, RFQI episodic reward stays at around 5500 whereas FQI drops faster to 4300 for more than 50% change in the nominal value. Similar robust performance of RFQI can be seen in other plots as well.

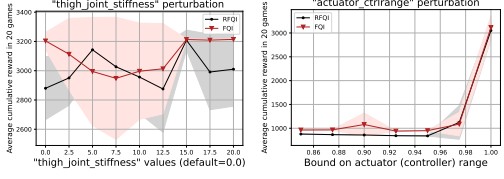

Figure 8: Similar performance of RFQI and FQI in Hopper on dataset $\mathcal{D}_d$ w.r.t. parameters *actuator_ctrlrange* and *thigh_joint_stiffness*.

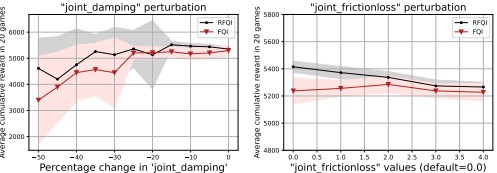

Figure 9: Similar performance of RFQI and FQI in Half-Cheetah on dataset $\mathcal{D}_d$ w.r.t. parameters *joint_damping* and *joint_frictionloss*.

As part of discussing the limitations of our work, we also provide two instances where RFQI and FQI algorithm behave similarly. RFQI and FQI algorithms trained on the D4RL dataset $\mathcal{D}_d$ perform similarly under the perturbations of the *Hopper* model parameters *actuator_ctrlrange* and *thigh_joint_stiffness* as shown in Fig. 8. We also make similar observations under the perturbations of the *Half-Cheetah* model parameters *joint_damping* (both front *joint_damping* and back *joint_damping*) and *joint_frictionloss* as shown in Fig. 9. We observed that the robustness performance can depend on the offline data available, which was also observed for non-robust offline RL algorithms (Liu et al., 2020; Fu et al., 2020; Levine et al., 2020). Also, perturbing some parameters may make the problem really hard especially if the data is not representative with respect to that parameter. We believe that this is the reason for the similar performance of RFQI and FQI w.r.t. some parameters. We believe that this opens up an exciting area of research on developing online policy gradient algorithms for robust RL, which may be able to overcome the restriction and challenges due to offline data. We plan to pursue this goal in our future work.