# OpenReview forum: "Robust Reinforcement Learning using Offline Data"
_NeurIPS.cc/2022/Conference — NeurIPS 2022 Accept_

### Official Review · Reviewer_3j8B · 2022-07-07

**Rating:** 5
**Confidence:** 3
**Soundness:** 3 good
**Presentation:** 3 good
**Contribution:** 3 good

**Summary:**

This paper endeavors to design a robust algorithm in the offline RL setting with function approximation. The proposed algorithm is called Robust Fitted Q-Iteration, hopefully addressing challenges in this specific setting. The detailed derivation of the proposal is given under certain assumptions, followed by a near-optimal guarantee or sample complexity result. Experiments are conducted on three typical environments.



**Questions:**

1.In Eq.10, I am wondering why seeking the optimal dual function $g$ is equivalent to an optimal dual variable, and what the approximation gap between $\hat{g_{f^\prime}}$ and $g^*_{f^\prime}$?

2.The paper [1] also considers the robust RL setting, so what is the difference between it with this paper? Relevant discussion should be added in the related work.

3.Three challenges mentioned in Section 1 seems only to involve Robust RL. If additionally considering the offline setting, is there any new challenge for offline robust RL?

[1] Corruption-Robust Offline Reinforcement Learning (AISTAT 2022)


**Limitations:**

1.The paper designs the robust algorithm only based on Total variation, which might be limited in terms of practice. I am not sure why the authors chose this distance rather than KL or others? Is TV the most typical choice in robust RL literature? An explanation is expected in the future.

2.The empirical improvement over other baselines seems not to be significant. More experiments are needed, including on more games. Also, a detailed comparison of the computational cost is also suggested.

In summary, I may not be familiar with some technical details, but as far as I can tell, they are basically sound. In particular, the dual reformulation as well as the following assumptions serves as the key to the proposed algorithm, which is the critical contribution of this paper. Therefore, I think it is a borderline paper as the algorithmic contribution is notable, while the practicability of the proposed algorithms might be questionable and needs to be further demonstrated.


**Strengths And Weaknesses:**

### Strength.
1.Related works and preliminaries are thorough and comprehensive.

2.The paper is well organized and easy to follow.


### Weakness
1.In the design process of Robust Fitted Q-iteration, the sequentially added assumptions, e.g., Assumptions 3 and 4, lack demonstration empirically. In practice, we are not fully sure whether these assumptions can be satisfied or not, and how they impact on empirical performance.

2.Emprical demonstration is insufficient as authors only compare their algorithm with a limited number of baselines, e.g., SR-DQN, on only two or three Mujoco games. Researchers tend to be concerned about whether the empirical superior performance of the proposed algorithm can be observed as well on other games.

3.The proof sketch might be wordy and hard to understand.

---

> ### Author Response · Authors · 2022-07-31
> **Author Response**
>
> We thank the reviewer for their thoughtful comments on our paper. We are encouraged by the fact that they found our paper well-organized, easy to follow, thorough, and comprehensive.  We address the major comments below and in our common response above. We will revise our paper based on these comments.
>
>
>
> > **Q1.** *''In practice, we are not fully sure whether these assumptions can be satisfied or not, and how they impact on empirical performance.''*
>
> **Response:** Thank you for the comment regarding the assumptions of the paper and for providing an opportunity for us to clarify.  Please see our general response **[CQ.3]**. Also, please see our response to Reviewer NXv3 for a more detailed explanation of each of the four assumptions we used in our paper.
>
>
> > **Q2.** *''Empirical demonstration is insufficient as authors only compare their algorithm with a limited number of baselines, e.g., SR-DQN, on only two or three Mujoco games.''*
>
> **Response:** Thank you for your comment. Please see our general response **[CQ.1]** above.
>
>
> >**Q3.** *''In Eq.10, I am wondering why seeking the optimal dual function $g$ is equivalent to an optimal dual variable, and what the approximation gap between $\hat{g}_{f'}$ and $g^*_{f'}$?.''*
>
> **Response:** As we have mentioned in lines 298-300, the correct dual variable function to be used in Eq.10 is the optimal dual variable  $g^*_{f'} = \arg\min_{g \in \mathcal{G}} L_{\mathrm{dual}}(g; f')$ corresponding to the last iterate $f'$. However, it is not possible to perform this optimization as $L_{\mathrm{dual}}$ involves taking an expectation w.r.t. $P^{o}$ which is unknown. So, we use the empirical risk minimization approach to replace the true expectation with sample average, and approximate $g^*_{f'}$  by  $\hat{g_{f'}} = \arg\min_{g \in \mathcal{G}} \hat{L_{\mathrm{dual}}}(g; f')$.  Note that $\hat{L_{\mathrm{dual}}}$ is a sample based approximation of $L_{\mathrm{dual}}$. In the technical analysis of our algorithm, we account for the approximation error from this step. Please see the proof sketch given in Section 4.4 for a brief discussion on how to bound this approximation error, and please see Lemma 6 in Appendix C  for more technical details.
>
>
> >**Q4.** *''The paper [Wang et. al., 2022] also considers the robust RL setting, so what is the difference between it with this paper?''*
>
> **Response:** Thank you for your comment. Please see our general response **[CQ.2]** above.
>
>
> >**Q5.** *''The paper designs the robust algorithm only based on Total variation, which might be limited in terms of practice. I am not sure why the authors chose this distance rather than KL or others? ''*
>
> **Response:** Since this is the first work on offline robust RL, we decided to focus on one specific uncertainty set and overcome the fundamental challenges and obtain the first of its kind theoretical result. Since the total variation distance is a popular distance metric used in supervised learning and in RL, we decided to consider it first. **Even with just considering the total variation distance, coming up with the dual formulation and its approximation, and combining it with action-value function theory were challenging research problems with many failed attempts and the final success which reached the form of this manuscript.**
>
> Having said that, we appreciate the reviewer's comment about considering other uncertainty sets. We have already mentioned this as one of our key future research directions in Section 6 (Conclusion) of the paper.

---

> > ### Comment · Reviewer_3j8B · 2022-08-07
> > **Thanks for the reply**
> >
> > I appreciate the authors' reply. I keep my rating as the response clarified some of my concerns. I am still not fully convinced by the practicability/validity of these assumptions this paper used, as well as the choice of TV, although it is the first robust offline RL paper as the authors mentioned.

---

> > > ### Author Response · Authors · 2022-08-08
> > > **Second Author Response**
> > >
> > > **Q6.** *``I am still not fully convinced by the practicability/validity of these assumptions this paper used, as well as the choice of TV, although it is the first robust offline RL paper as the authors mentioned.''*
> > >
> > > **Response:** We thank the reviewer for their reply to our initial response.
> > >
> > > We take this opportunity to further discuss the topic of *practicality of assumptions*. We note that we will include these discussions in the camera-ready version of our manuscript. **Assumptions 2 and 4** are about the realizability of the optimal solutions which are typically fulfilled by the representation power of the deep neural networks. **Assumption 3** is about the existence of a *fail-state*. We have provided its practicality in lines 228-230 of our manuscript and we have further discussed its necessity in our response to Reviewer NXv3 at **Q4**. To provide further examples, we note that in the MuJoCo environments like Hopper, Half-cheetah when the robot falls down or goes out of bounds in the world, no matter what action is taken we cannot lift it up, and such a state is a fail-state. Finally, **Assumption 1** is concentratability which is now a standard assumption in offline RL theory (Munos, 2003; Agarwal et al., 2019; Chen and Jiang, 2019; Wang et al., 2021; Xie et al., 2021). In fact, it is necessary to have this concentratability assumption (or a variation of this) in order to show the provable optimality of offline RL algorithms (see Chen and Jiang, 2019; Wang et al., 2021; and multiple references therein). While it is typically difficult to verify the satisfiability of this assumption empirically from the given data alone, this assumption intuitively asks for *good coverage* of the data that  spans the state  and action spaces as much as possible.  The D4RL benchmark datasets (Fu et al., 2020; Levine et al., 2020) are based on this intuition which provides a variety of datasets such as *random dataset* (generated by the uniform data distribution), *expert dataset* (generated by a trained RL agent), *medium dataset* (mixture of random and expert), and many more such combinations. The performance of the policies obtained from these offline datasets are inherently connected to the quality of these datasets, as shown in the D4RL paper (Fu et al., 2020; Levine et al., 2020).
> > >
> > > *On the choice of TV.* We would like to further add details to our response in **Q5** above. We mentioned that even with just the TV distance, coming up with the dual formulation and its approximation, and combining it with action-value function theory were challenging research problems with many failed attempts and the final success which reached the form of this manuscript. Unfortunately, for the other metrics like chi-square and KL, we found their dual formulations to be non-linear in the $\mathbb{E}_{P^o}[.]$ term which makes it hard to come up with the least-squares solution like eq.(10). This is still currently work-in-progress by our group and hence we deferred it to our future work as mentioned in our conclusion. Nonetheless, TV distance analysis and practical algorithm came with its own challenges that we are really proud of solving and to present in this manuscript (9 pages main paper and a self-contained detailed theory and practical algorithm in the appendix).
> > >
> > >
> > > We are thankful to the reviewer for their fact-check positive comment: *``it is the first robust offline RL paper as the authors mentioned.''*  We hope that the reviewer is satisfied with our response and we are happy to further clarify/discuss any other comments. We sincerely hope that the reviewer rating can be adjusted to reflect the detailed response and the reviewer's positive comment above.

---

### Official Review · Reviewer_NZk6 · 2022-07-08

**Rating:** 6
**Confidence:** 3
**Soundness:** 2 fair
**Presentation:** 2 fair
**Contribution:** 2 fair

**Summary:**

The paper deals with robust offline RL using function approximators. Robustness is a broad term, here it means a special robustness defined in the RMDP framework. An algorithm is outlined and numerous proofs of the properties of the algorithm are proved under strong assumptions. The algorithm is evaluated on two deterministic benchmarks (cart pole and hopper).
Robustness to changes in system dynamics is investigated, and promising performance is achieved on the two selected benchmarks relative to the selected comparison algorithms.


**Questions:**

The proofs make several assumptions, it would be interesting to see to what extent these assumptions are met in practice. Specifically, it would be interesting to see how often it happens in the experiments that the assumptions of the proofs are fulfilled.

In Figure 1 and 2 the algorithms FQI, DQN, SR-DQN and RFQI are examined, in Figure 3 it is FQI, TD3, SRDDPG and RQI. Why were the same algorithms not compared on all benchmarks? In addition, the same colors are used in the three figures to identify the algorithms, although they are not the same algorithms. This should be explained or changed.

In Figure 3, it is noticeable that for small values of "lag_joint_stiffness" of 0, 5 and 10 TD3 is clearly superior to the proposed RFQI. This should be mentioned in the text. Possibly it should be mentioned as a limitation. Also, it should be explained why this does not contradict claim „we presented a novel robust RL algorithm called Robust Fitted Q-Iteration algorithm with provably optimal performance for an RMDP with arbitrarily large state space, using only offline data with function approximation“. Furthermore, claim „We also demonstrated the superior performance of the proposed algorithm on standard benchmark problems“ seems to be an exaggeration.

„Parameter uncertainty commonly occurs in many real-world RL applications due to simulator modeling errors, changes in the real-world system dynamics over time, and ad\ersarial disturbances“\
I am skeptical that "Parameter uncertainty commonly occurs in many real-world RL applications due to ... adversarial disturbances". I don't think you can write it that way without giving a reference to the source of this assessment.


Lesser importance:\
„Robust RL is typically formulated as a max-min problem“\
I would rather say this is one variant, others use percentile performance optimization.

"Learning policies for RL problems with large state-action spaces is computationally intractable."\
This claim is not correct in this form (backgammon, go, chess, manifold applications with continuous state and action spaces).

In the bibliography there are a few wrong lower case letters, like: bayesian, mdp





**Limitations:**

There are, in my opinion, a few limitations that should still be addressed in the text.

To what extent the conditions for the validity of the proofs are realized in practice should be discussed as a possible limitation.

Furthermore, the limitation that the benchmarks used have deterministic dynamics throughout should be mentioned.

**Strengths And Weaknesses:**

**Strengths**
* The paper makes extensive theoretical contributions.
* The proposed practical algorithm could be promising.

**Weaknesses**
* The paper makes exaggerated claims.\
"In this work, we presented a novel robust RL algorithm called Robust Fitted Q-Iteration algorithm
with provably optimal performance for an RMDP with arbitrarily large state space, using only offline data with function approximation."\
Optimal? Provably optimal? The limitations and assumptions should be named. The non-optimality evident in Figure 3 should be explained.\
„We also demonstrated the superior performance of the proposed algorithm on standard benchmark problems.“\
Based on figures 1, 2 and 3, there is too little evidence, that the performance is superior. Furthermore, the statement can only be made for the examined alternative algorithms, furthermore there are only two benchmarks.

* The empirical evidence of the usefulness of the method is weak.\
Only two benchmarks, moreover both contain only deterministic dynamics. Due to the lack of investigations on benchmarks with stochastic dynamics, it seems questionable to me whether the experiments are at all suitable to test the claim of the proposed method.
* The paper seems incomplete without the Appendix.\
Algotithm 2. More experiments.

---

> ### Author Response · Authors · 2022-07-31
> **Author Response - part 1/2**
>
> We thank the reviewer for their thoughtful comments on our paper. We are encouraged by the fact that they found our theoretical contributions extensive.  We address the major comments below and in our common response above. We will revise our paper based on these comments.
>
> >**Q1.** *''The non-optimality evident in Figure 3 should be explained.''  ''In Figure 3, it is noticeable that for small values of "leg\_joint\_stiffness" of 0, 5 and 10 TD3 is clearly superior to the proposed RFQI. \ldots Also, it should be explained why this does not contradict claim ''we presented a novel robust RL algorithm called Robust Fitted Q-Iteration algorithm with provably optimal performance for an RMDP with arbitrarily large state space, using only offline data with function approximation". Furthermore, claim ''We also demonstrated the superior performance of the proposed algorithm on standard benchmark problems" seems to be an exaggeration.'' ''*
>
> **Response:** Thank you very much for your question. There seems to be a misunderstanding about the difference between the notion of the terms *optimality* and the *superior performance* in the robust RL and non-robust RL contexts. We agree that this might be confusing to readers who are in general unfamiliar with the robust RL literature. We are sorry for this confusion and we will add a clear description in our revised manuscript to avoid any such possible confusion.
>
> Here, "optimality for an RMDP" is referring to optimal robust value function and/or optimal robust policy that specifically correspond to the RMDP framework. In Section 2 (Preliminaries), we formally state: "the RMDP problem is to find the *optimal robust policy* which maximizes the value against the worst possible model in the uncertainty set $\mathcal{P}$...and *the optimal robust value function* $V^*$ are defined as $V^* = \sup_{\pi}\inf_{P\in\mathcal{P}} V_{\pi,P}$". Different from this, a policy $\bar{\pi}^*$ is optimal in the non-robust MDP setting if it maximizes the value function defined w.r.t. to the model $P^{o}$ of that MDP, i.e, $\bar{\pi}^{*} = \sup_{\pi} V_{\pi, P^{o}}$. Please note that $\inf_{P \in \mathcal{P}}$ is absent in this non-robust optimality definition.
>
> RFQI is superior/optimal compared to TD3 in terms of robustness (as defined in the RMDP framework) instead of performance merely at the nominal model $P^{o}$. The goal of robust RL is not to optimize performance with respect to *one model* but to show tenacity, or reluctance to drop performance, under a much wider range of model perturbations. This idea also manifests precisely in the mathematical formulation of the RMDP framework in that we are optimizing the policy with respect to the most adversarial model given an uncertainty set. In contrast, most non-robust RL algorithms aim to perfect performance with respect to a given model which, in our case, is the nominal model, and which, in reference to Figure 3, is the small values of "leg\_joint\_stiffness" of 0, 5, and 10 as the reviewer mentioned. Thus, it is expected that TD3 shows better performance near the nominal model. Figure 3 clearly shows that TD3 has a catastrophic drop in performance near "leg\_joint\_stiffness=20" while RFQI outperforms everyone in terms of robustness. In particular, for more than $30\%$ change in the leg\_joint\_stiffness parameter, the episodic reward of RFQI is around 1800, which is significantly more than that of TD3 which gets an episodic reward of only around 300. That is, RFQI maintains reasonable performance under prolonged and more serious model perturbation.
>
> Again, we thank you for bringing up this point since it relates to our central argument. If one is very accustomed to thinking "optimality" only in non-robust MDP, confusion naturally arises. We acknowledge this confusion, and we will emphasize it in our revised manuscript in order to facilitate readers adapting to notions in the robust RL literature.
>
> ---
> *Please see the next comment for continuation.*

---

> > ### Author Response · Authors · 2022-07-31
> > **Author Response - part 2/2**
> >
> >
> > *Continued ...*
> >
> > >**Q2.** *''In Figure 1 and 2 the algorithms FQI, DQN, SR-DQN and RFQI are examined, in Figure 3 it is FQI, TD3, SRDDPG and RQI. Why were the same algorithms not compared on all benchmarks? In addition, the same colors are used in the three figures to identify the algorithms, although they are not the same algorithms. This should be explained or changed.''*
> >
> > **Response:** Thank you for pointing this out. Please first note that CartPole is a discrete-action environment while Hopper has a continuous action space. DQN is a canonical value based learning method that only handles discrete action space. DDPG is an actor-critic method that extends from DQN and handles continuous action space as well. They both use temporal difference and experience replay. Thus we carefully pick SR-DQN and SR-DDPG with the intention that they represent similar methodology in different trenches of control problems. As for the non-robust benchmark, we choose TD3 since it is the successor to DDPG  and a **state-of-art** algorithm for continuous control problems. We agree with the reviewer that a more careful explanation should be offered here to convey the intention. We will add this explanation in the revised manuscript.
> >
> > >**Q3.** *''The proofs make several assumptions ...''*
> >
> > **Response:** Thank you for the comment regarding the assumptions of the paper and for providing an opportunity for us to clarify.  Please see our general response **[CQ.3]**. Also, please see our response to Reviewer NXv3 for a more detailed explanation of each of the four assumptions we used in our paper.
> >
> > >**Q4.** *''I am skeptical that "Parameter uncertainty commonly occurs in many real-world RL applications due to ... adversarial disturbances". I don't think you can write it that way without giving a reference to the source of this assessment''*
> >
> > **Response:** We thank the reviewer for pointing this out. One of our key references to make this statement was [R1] (and multiple references therein) which uses a robust MDP formulation to handle adversarial perturbations. In our revised version, we will add more citations to support this statement.
> >
> > [R1] Michael Lutter, Shie Mannor, Jan Peters, Dieter Fox, and Animesh Garg. ``Robust value iteration for continuous control tasks.'', arXiv preprint arXiv:2105.12189 (2021).
> >
> > >**Q5.** *$(i)$.  ''Robust RL is typically formulated as a max-min problem". I would rather say this is one variant, others use percentile performance optimization''. $(ii)$ ''Learning policies for RL problems with large state-action spaces is computationally intractable." This claim is not correct in this form (backgammon, go, chess, manifold applications with continuous state and action spaces).''*
> >
> > **Response:** Thank you for pointing out these. We will fix $(i)$ as  ''One of the main approaches for solving the robust RL problem is through a max-min formulation'' and $(ii)$ as ''Learning policies for RL problems with large state-action spaces is computationally intractable *in the tabular setting*.''

---

> > > ### Comment · Reviewer_NZk6 · 2022-08-07
> > > **Fulfillment of assumptions in experiments**
> > >
> > >
> > > Thank you for the answers and promised changes,
> > >
> > > they clarify some questions and will avoid some misunderstandings.
> > > However, I am still very interested in an answer to
> > >
> > > `Specifically, it would be interesting to see how often it happens in the experiments that the assumptions of the proofs are fulfilled.`
> > >
> > > and even if the result is that the assumptions are never fulfilled in the experiments, that is not a compelling reason to reject the paper My observation is that it happens in many papers that in the experiments the assumptions of the thoretic consideration are not fulfilled. But I think it is important that authors and readers are aware of this situation, if it exists.

---

> > > > ### Author Response · Authors · 2022-08-08
> > > > **Second Author Response**
> > > >
> > > > **Q6.** *``Specifically, it would be interesting to see how often it happens in the experiments that the assumptions of the proofs are fulfilled.''*
> > > >
> > > > **Response:** We thank the reviewer for their reply to our initial response.
> > > >
> > > > We take this opportunity to further discuss the topic of *practicality of assumptions*. We note that we will include these discussions in the camera-ready version of our manuscript. **Assumptions 2 and 4** are about the realizability of the optimal solutions which are typically fulfilled by the representation power of the deep neural networks. **Assumption 3** is about the existence of a *fail-state*. We have provided its practicality in lines 228-230 of our manuscript and we have further discussed its necessity in our response to Reviewer NXv3 at **Q4**. To provide further examples, we note that in the MuJoCo environments like Hopper, Half-cheetah when the robot falls down or goes out of bounds in the world, no matter what action is taken we cannot lift it up, and such a state is a fail-state. Finally, **Assumption 1** is concentratability which is now a standard assumption in offline RL theory (Munos, 2003; Agarwal et al., 2019; Chen and Jiang, 2019; Wang et al., 2021; Xie et al., 2021). In fact, it is necessary to have this concentratability assumption (or a variation of this) in order to show the provable optimality of offline RL
> > > > algorithms (see Chen and Jiang, 2019; Wang et al., 2021; and multiple references therein). While it is typically difficult to verify the satisfiability of this assumption empirically from the given data alone, this assumption intuitively asks for *good coverage* of the data that  spans the state  and action spaces as much as possible.  The D4RL benchmark datasets (Fu et al., 2020; Levine et al., 2020) are based on this intuition which provides a variety of datasets such as *random dataset* (generated by the uniform data distribution), *expert dataset* (generated by a trained RL agent), *medium dataset* (mixture of random and expert), and many more such combinations. The performance of the policies obtained from these offline datasets are inherently connected to the quality of these datasets, as shown in the D4RL paper (Fu et al., 2020; Levine et al., 2020).
> > > >
> > > >
> > > > **Q7.** *``and even if the result is that the assumptions are never fulfilled in the experiments, that is not a compelling reason to reject the paper''*
> > > >
> > > > **Response:** We are thankful of the reviewer for this positive comment.  We hope that the reviewer is satisfied with our response and we are happy to further clarify/discuss any other comments. We sincerely hope that the reviewer rating can be adjusted to reflect the detailed response and the reviewer's positive comment above.

---

> > > > > ### Comment · Reviewer_NZk6 · 2022-08-09
> > > > > **Further remark**
> > > > >
> > > > > Thank you very much for this detailed response, I am seeing your work less critical now (I wll update my score in the reviewer-AC discussion round).
> > > > >
> > > > > One further remark that I want to place in this discussion in the hope, that it might help the scientific progress:
> > > > >
> > > > > You state:
> > > > > ` it is typ,ically difficult to verify the satisfiability of this assumption empirically from the given data alone`
> > > > >
> > > > > I find the situation unsatisfactory that the conditions for the validity of the assumptions seem to be untestable in practice.
> > > > >
> > > > > ` this assumption intuitively asks for good coverage of the data that spans the state and action spaces`
> > > > >
> > > > > Unfortunately, this good coverage is something that is rarely achieved, even a random policy does not guarantee it, provided the amount of data is limited. While all actions are tried in the frequently used states, typically parts of the state space are not reached at all or only very rarely.
> > > > >
> > > > > However, I do not see this at all, as a problem to be solved in this paper. It affects a whole class of theoretical work. I just wanted to use this discussion to point out this problem. As an example in [1] the requirement for data coverage are clearly defined (there it is simple, because it is about finite discrete MDPs. And [1] is about a different topic, but the point I want to make is that the prerequisite concerning data coverage is expressed in an manner, that it can be tested), however the requirement so defined are not met in the experiments (see e.g. [2] for a discussion).
> > > > >
> > > > > Please consider this comment really only as a contribution to the discussion and a suggestion for the future, I do not in any way blame your work for this, as I see it,  general shortcoming.
> > > > >
> > > > > [1] K. Nadjahi et al, Safe Policy Improvement with Soft Baseline Bootstrapping
> > > > >
> > > > > [2] P. Scholl et al., Safe Policy Improvement Approaches on Discrete Markov Decision Processes

---

### Official Review · Reviewer_NXv3 · 2022-07-09

**Rating:** 5
**Confidence:** 3
**Soundness:** 2 fair
**Presentation:** 3 good
**Contribution:** 3 good

**Summary:**

This paper suggests a method to solve robust MDP using offline data. A fitted Q-learning is proposed and the authors developed its convergence and sample complexity.

**Questions:**

(1). See Weakness above.
(2). Assumption 4 assumes the representative ablity of the function class, can the authors provide some example? Linear function approximation? Neural network?
(3) Assumption 1 assumes an upper bound of $\mu$, and hence we have $\min_{s,a} \mu(s,a)>0$. As the number of samples increases, does this means we are getting samples of all $\mathcal{S}\times\mathcal{A}$? So does this situation in some sense recover the generated model setting (for any policy $\pi$, the offline data can be viewed as a generated sample)?

**Limitations:**

This paper is well-written, but some of the assumptions seem to be a little strong to me and hence limit the contribution of the paper.

**Strengths And Weaknesses:**

S: This work considered the RL problems under the offline setting, which is more challenging. A global optimality is also proposed.

W: (1).The main challenge in robust RL is to find the worst case, however in this paper, it is solved by assuming a fail-state. This assumption greatly reduces the difficulties of the problem hence I'm afread it is too strong. Although it can be justified in practice, it is still too strong for me.

---

> ### Author Response · Authors · 2022-07-31
> **Author Response - part 1/2**
>
> We thank the reviewer for their thoughtful comments on our paper. We are encouraged by the fact that they found our paper addresses a challenging problem and presents the global optimality result. Thank you also for commending that our paper is well-written.  We address the major comments below and in our common response above. We will revise our paper based on these comments.
>
> >**Q1.** *``This paper is well-written, but some of the assumptions seem to be a little strong to me and hence limit the contribution of the paper''*
>
> **Response:** Thank you for the comment regarding the assumptions of the paper and for providing an opportunity for us to clarify. We use four main assumptions in the paper. Assumption 1 and Assumption 2 are standard assumptions widely used in the offline RL literature (please see our response to **Q2** below). Assumption 4 is similar to the realizability assumption in the offline RL (please see our response to **Q3** below). Assumption 3 is a technical assumption employed to make the algorithm more computationally tractable and it is unavoidable for the TV uncertainty set (please see our response to **Q4** below). We sincerely believe that these assumptions are minimal and standard in the context of the existing literature on the offline RL theory. Moreover, they do not affect the validity/applicability of the RFQI algorithm we developed as evident from the simulation results we presented in Section 5 and Appendix E. In our revised manuscript, we will give a more detailed justification for using these assumptions to address the concerns of the reviewer and other readers.
>
> >**Q2.** *``Assumption 1 assumes an upper bound of $\mu$ , and hence we have $\min_{s,a}\mu(s,a) > 0$. As the number of samples increases, does this means we are getting samples of all $\mathcal{S} \times \mathcal{A}$? So does this situation in some sense recover the generated model setting (for any policy $\pi$, the offline data can be viewed as a generated sample)''*
>
> **Response:** Assumption 1 is called the concentratability assumption in the offline RL literature. This is a standard assumption widely used in the offline RL theory literature (Munos, 2003; Agarwal et al., 2019; Chen and Jiang, 2019; Wang et al., 2021; Xie et al., 2021). In fact, it is necessary to have this concentratability assumption (or a variation of this) in order to show the provable optimality of offline RL algorithms (see Chen and Jiang, 2019; Wang et al., 2021; and multiple references therein). Since our robust offline RL algorithm builds on the framework of the standard non-robust offline RL algorithm, we also used this standard assumption.
>
> With respect to the specific question by the reviewer, it is indeed true that as the number of samples increases to infinity, the probability of getting a sample corresponding to any arbitrary $(s,a)$ pair will also increase to one. However, this will not be identical to the generative model setting because the number of samples corresponding to different $(s,a)$ pairs will still be different. In the generative model based sample complexity analysis of the RL algorithms (Azar et al., 2013; Sidford et al., 2018; Agarwal et al., 2020; Li et al., 2020; Kalathil et al., 2021; Panaganti and Kalathil, 2022), it is typically assumed that the data contains the same number of samples corresponding to each $(s,a)$ pair.
>
> >**Q3.** *``Assumption 4 assumes the representative ability of the function class, can the authors provide some example?''*
>
> **Response:** As the reviewer observed, neural networks can satisfy this assumption on $\mathcal{G}$ due to their representation power. However,  $\mathcal{G}$ can be other function classes also, depending on $\mathcal{F}$ and $L_{\mathrm{dual}}$. In particular, if $\mathcal{F}$ is finite $\mathcal{G}$ can also be finite, which is the case we are considering in this paper. We would like to point out that Assumption 4 which specifies approximate dual realizability is similar to the standard realizability and completeness assumption in offline RL literature (Agarwal et al., 2019; Chen and Jiang, 2019; Wang et al., 2021; Xie et al., 2021), see Assumption 2.
>
> ---
>
> *Please see the next comment for continuation.*

---

> > ### Author Response · Authors · 2022-07-31
> > **Author Response - part 2/2**
> >
> >
> >
> > *Continued...*
> >
> > >**Q4.** *``The main challenge in robust RL is to find the worst case, however in this paper, it is solved by assuming a fail-state [Assumption 3]''*
> >
> > **Response:** Thank you for presenting your concern. As we have mentioned in lines 223-225, finding  $\inf_{s''} V(s'')$ is straightforward in a tabular setting (due to the assumption of working with finite state space), and the theory will go through without the fail-state assumption (see Panaganti and Kalathil, 2021).  However, finding  $\inf_{s''} V(s'')$  is infeasible in a function approximation setting. It is to overcome this *computational* challenge, we used the natural assumption of fail-state. We have clearly justified this assumption in the paper, please see lines 228-230. Moreover, since this $\inf_{s''} V(s'')$ term is inevitable in the dual formulation (given in Proposition 1),  it is not possible to avoid this assumption (or an equivalent assumption) for coming up with a computationally tractable algorithm.
> >
> > One possible way to avoid this fail-state assumption is to consider uncertainty sets such as KL and Chi-Square uncertainty sets, different from the TV uncertainty set considered in this paper. This is one of the key future work directions which we have mentioned in Section 6 of the paper.

---

### Official Review · Reviewer_9rit · 2022-07-13

**Rating:** 6
**Confidence:** 3
**Soundness:** 3 good
**Presentation:** 2 fair
**Contribution:** 3 good

**Summary:**

Offline RL has been receiving enormous attention recently due to its practical application. However, as the offline data generation method could be noisy, identifying a robust solution by accounting uncertain model parameters has great impact. The paper proposes a robust fitted Q-learning (RFQI) algorithm that used a dual formulation to work with nominal training model. The dual problem is reformulated with function optimization and solved with risk minimization techniques, which is then used to approximate robust Bellman update. Theoretical upper bounds are derived for the RFQI algorithm. Experimental results on simple toy environments demonstrate that RFQI outperforms non-robust algorithms.

**Questions:**

1. How is your work fundamentally different from R-LSVI?
2. Are you planning to open-source the codes for reproducibility purposes?


**Limitations:**

The paper missed citing some recent work on offline robust RL (e.g., Zhang et. al. 2022). Experimental results are not sufficient to empirically validate the claims. Validating the performance against state-of-the-art robust offline RL methods on a set of real-world simulated environments will strengthen the paper.

**Strengths And Weaknesses:**

Robust offline RL is a practically important and challenging problem. RFQI reformulated the challenging estimation problem to a function optimization model and used it to update the robust Q-function. I appreciate the theoretical results and insights provided in Theorem 1. Experiments are performed on simple gym environments which are not exhaustive enough. Furthermore, only non-robust benchmark algorithms are used to validate the performance. Comparing against something like R-LSVI (Zhang et. al. 2022) would have been great.

References:
1.	Zhang, Xuezhou, Yiding Chen, Xiaojin Zhu, and Wen Sun. "Corruption-robust offline reinforcement learning." In International Conference on Artificial Intelligence and Statistics, pp. 5757-5773. PMLR, 2022.

---

> ### Author Response · Authors · 2022-07-31
> **Author Response**
>
> We thank the reviewer for their thoughtful comments on our paper. We are encouraged by the fact that they found our theoretical results insightful.  We address the major comments below and in our common response above. We will revise our paper based on these comments.
>
> >**Q1.** *``How is your work fundamentally different from R-LSVI (Zhang et. al., 2022)?''*
>
> **Response:** Thank you for your question. Please see our general response **[CQ.2]** above.
>
> >**Q2.** *``Are you planning to open-source the codes for reproducibility purposes?''*
>
> **Response:** Yes, we are. Please note that in Section E of Appendix (Experiment Details) we have already included a link to our anonymous github repo which contains the code to reproduce the results in the paper. We plan to release a public codebase after the paper decision.
>
> >**Q3.** *``Experiments are performed on simple gym environments which are not exhaustive enough''.* *``Validating the performance against state-of-the-art robust offline RL methods on a set of real-world simulated environments will strengthen the paper. ''*
>
> **Response:** Thank you for your comment. Please see our general response **[CQ.1]** above.

---

### Author Response · Authors · 2022-07-31
**General author response to everyone - part 1/2**

We thank all the reviewers for their thoughtful comments and suggestions. We first give our response to the concerns raised by more than one reviewer as a single common response.

**[CQ.1] On the sufficiency/exhaustiveness of the simulation experiments**

**Response:**  We sincerely believe that our experimental evaluation validates the performance of our algorithm, by comparing it against non-robust RL algorithms (FQI and TD3) and two online robust RL algorithms (SR-DQN and SR-DDPG). In particular, kindly consider the following aspects:

$(i)$ As we have stated in our paper, *to the best of our knowledge, ours is the **first work** which proposes an **offline** RL algorithm to solve the robust MDP/RL problem.* In this context, to the best of our knowledge, *there are no other state-of-the-art **offline** robust   RL methods that we can compare against*. So, we chose to compare the performance against two *online* robust   RL algorithms, SR-DQN and SR-DDPG, see Fig.3 and Fig.4. Note that SR-DQN and SR-DDPG do not have any provable guarantees on the performance of the learned policy. Significantly differently from this, *we give (first of this kind) provable guarantee on the performance of our offline robust RL algorithm*.


$(ii)$ We have a detailed section on the experiments, in  Appendix E, in addition to the one page section in the main paper. In Appendix E.1, we have developed a practical implementation of our algorithm (Algorithm 2), specified the hyperparameters, described how we collected three different types of offline data, and specified the computing resources used. Appendix E.2 contains 12 additional plots (Fig.4 - Fig.7) which demonstrate the performance of our RFQI algorithm against its non-robust counterpart and a state-of-the-art non-robust RL algorithm (TD3 algorithm). These results clearly show the robust performance of the policy learned by our RFQI algorithm.


$(iii)$ We have demonstrated the performance of our RFQI algorithm in two challenging MuJoCo tasks (Hopper and Half-Cheetah) and one OpenAI Gym task  (CartPole). As mentioned in  Appendix E, we implemented our RFQI  algorithm based on the architecture of BCQ algorithm (Fujimoto  et al., 2019) and PQL algorithm (Liu et al., 2020). Note that both these non-robust offline RL algorithms are evaluated on standard MuJoCo tasks (Hopper, HalfCheetah, and Walker2d for BCQ;  CatPole and Hopper for PQL). In our paper, we address the much more challenging robust RL problem, and still provide detailed  experimental evaluation on these standard MuJoCo tasks. Please note that in Section E of Appendix we have included a link to our anonymous github repo which contains the code to reproduce the results in the paper.

Considering that the offline robust RL problem is significantly more challenging than vanilla offline RL, we think that the current number of simulations should by no means diminish the **value of our work as the first theoretical paper in the area of robust offline RL**. We sincerely believe that this paper contributes to expanding the theoretical knowledge in the relatively new and challenging area of robust RL.

**[CQ.2] On the similarities/differences with the R-LSVI algorithm (Zhang et. al., 2022)**

(Zhang et. al., 2022) Xuezhou Zhang, Yiding Chen, Xiaojin Zhu, and Wen Sun. `Corruption-robust offline reinforcement learning', International Conference on Artificial Intelligence and Statistics, 2022.

**Response:** Our work is fundamentally different from that of (Zhang et. al., 2022), in terms of the basic problem addressed, algorithms developed, and the technical analysis used.

(Zhang et. al., 2022) addresses the problem of \textit{corruption-robust} offline RL problem, where an adversary is allowed to change a fraction of the samples of an offline RL data set and the goal is to find the optimal policy for the **nominal MDP model** (according to which the offline data is generated). This problem addressed in (Zhang et. al., 2022) is fundamentally different from that of the *Robust Markov Decision Process (RMDP)* formulation (Iyengar, 2005; Nilim and El Ghaoui, 2005) where the goal is to find the optimal robust policy that performs the best under the worst possible MDP model in an uncertainty set. In our paper, we address the problem of finding an offline RL algorithm with provable performance guarantees for this RMDP problem. Note that (Zhang et. al., 2022) do not even cite any of the standard RMDP literature and the RL approaches to solve the RMDP problem. So, the basic problems addressed in these two papers are completely different. Also note that (Zhang et. al., 2022) is restricted to a linear MDP setting and do not provide any experimental results on the performance of their algorithm.

We apologize for missing this reference in our literature review. We will include this work and other related works on corruption robust RL in the literature review of our revised manuscript.

---
*Part 1 of 2*

---

> ### Author Response · Authors · 2022-07-31
> **General author response to everyone - part 2/2**
>
> **[CQ.3] On the validity of the assumptions**
>
> **Response:** Thank you for the comment regarding the assumptions of the paper and for providing an opportunity for us to clarify. We use four main assumptions in the paper. Assumption 1 and Assumption 2 are standard assumptions widely used in the offline RL literature. Assumption 4 is similar to the realizability assumption in the offline RL. Assumption 3 is a technical assumption employed to make the algorithm more computationally tractable and it is unavoidable for the TV uncertainty set. We sincerely believe that these assumptions are minimal and standard in the context of the existing literature on the offline RL theory. Moreover, they do not affect the validity/applicability of the RFQI algorithm we developed as evident from the simulation results we presented in Section 5 and Appendix E. In our revised manuscript, we will give a more detailed justification for using these assumptions to address the concerns of the reviewer and other readers.
>
> ---
> *Part 2 of 2*

---

### Meta-Review · Area_Chair_BkcM · 2022-08-26

**Recommendation:** Accept
**Confidence:** Certain

**Metareview:**

The reviewers are in general positive about the paper. The main concerns were about the assumptions used in the analysis. The AC is satisfied by the response from authors, and also thinks the assumptions are reasonable (standard in offline RL literature). The AC also thinks the setting studied in this paper is important.

**Award:**

No

---

### Decision · Program_Chairs · 2022-09-14

Accept